# THE KOLMOGOROV TEST: COMPRESSION BY CODE GENERATION

**Ori Yoran**[1,2]\*, **Kunhao Zheng**[1], **Fabian Gloeckle**[1]
**Jonas Gehring**[1], **Gabriel Synnaeve**[1], **Taco Cohen**[1]
[1]Meta AI (FAIR), [2]Tel Aviv University
`ori.yoran@cs.tau.ac.il`
`{kunhao, fgloeckle, jgehring, gab, tscohen}@meta.com`

## ABSTRACT

Compression is at the heart of intelligence. A theoretically optimal way to compress any sequence of data is to find the shortest program that outputs that sequence and then halts. However, such *Kolmogorov compression* is uncomputable, and code generating LLMs struggle to approximate this theoretical ideal, as it requires reasoning, planning and search capabilities beyond those of current models. In this work, we introduce the KOLMOGOROV-TEST (KT), a compression-as-intelligence test for code generating LLMs. In KT a model is presented with a sequence of data at inference time, and asked to generate the shortest program that produces the sequence. We identify several benefits of KT for both evaluation and training: an essentially infinite number of problem instances of varying difficulty is readily available, strong baselines already exist, the evaluation metric (compression) cannot be gamed, and pretraining data contamination is highly unlikely. To evaluate current models, we use audio, text, and DNA data, as well as sequences produced by random synthetic programs. Current flagship models perform poorly – both GPT4-O and LLAMA-3.1-405B struggle on our natural and synthetic sequences. On our synthetic distribution, we are able to train code generation models with lower compression rates than previous approaches. Moreover, we show that gains on synthetic data generalize poorly to real data, suggesting that new innovations are necessary for additional gains on KT.

## 1 INTRODUCTION

Compression and code generation are deeply related through the notion of Kolmogorov complexity, denoted $K(x)$, which is defined as the length of the shortest computer program[1] that produces the sequence $x$ as output and hence constitutes the optimal compression of $x$ (Kolmogorov, 1963; Li & Vitányi, 1997; Hutter et al., 2024) (see §2 for a detailed background). Kolmogorov complexity is *uncomputable* as it reduces to the halting problem, making the search for improved computable *upper bounds* a never-ending challenge and a potential benchmark for intelligence that by definition cannot saturate. We propose to view code generation language models (CODELMS) as upper bounds for Kolmogorov complexity: we task them to identify patterns in an input sequence and to compress it by producing a short program that outputs said sequence, and can measure their accuracy at producing correct programs as well as the compression rates achieved. This is the KOLMOGOROV-TEST (KT), a benchmark for reasoning capabilities for CODELMS, illustrated in Fig. 1.

We point out the following benefits of compression by code generation as a benchmark for the reasoning capabilities of language models: (1) the compression metric can be trusted in that it does not produce false positives, (2) diverse and richly structured sequence data is abundantly available, (3) it is highly unlikely that pretrained models have seen many relevant (program, sequence) pairs, making memorization-based solutions infeasible[2], (4) if the benchmark is saturated, either through improved reasoning and search capabilities or memorization, we can simply increase the sequence

---

\*Work done during an internship at Meta FAIR.
[1]For some universal Turing machine $U$.
[2]Models trained on synthetic data or by RL could very well end up using memorization.

Figure 1: **Data compression by code generation.** Consider compressing a sequence of bytes (presented as numbers in range $[0, 255]$) that can be produced by composing simpler sub-sequences. Standard compression methods, such as GZIP, focous on repetitions and frequency of characters and fail to exploit the logical patterns in this sequence (although they are strong baselines for long sequences, §5.3). LLMs are better at finding complex patterns, such as a sequence of incremental numbers, and can be used for compression with arithmetic coding. However, they are sensitive to phase-shifts due to their auto-regressive manner, and require model weights for decoding. Code generative models, inspired by the concept of Kolmogorov Complexity, can identify patterns in the input sequence to generate concise programs whose execution produces the original sequence.

length and (5) following research spanning decades of theoretical computer science, classical compression algorithms such as GZIP (Deutsch, 1996) can serve as strong baselines.

To evaluate current CODELMS on KT, we use naturally occurring sequences from three data modalities: text, audio, and DNA (§3.2). As we do not know the optimal program for the sequences, it is not possible to use this data for supervised learning, making it ideal for evaluation. To collect program-sequence pairs for supervised training and evaluation, we design a *compositional* domain-specific language (DSL) coupled with an automatic data generation framework (§3.3), inspired by context-free grammars (Chomsky, 1956; Hopcroft et al., 2006).

Our experiments (§4) show that sequence compression is an extremely challenging task for current CODELMS. Strong LLAMA-3.1-405B (Grattafiori et al., 2024) and GPT4-O (OpenAI et al., 2024b) models generate Python programs that fail to produce the input sequence in $78\%$ and $40\%$ of the time for naturally occurring data, and $66\%$ and $45\%$ percent of the time for synthetic sequences that follow clear patterns. On our synthetic distribution, we are able to train relatively small CODELMS with 1.5B parameters that outperform state-of-the-art prompted models by more than $40\%$, but struggle on real sequences. We further find that the prior used for program-compression is an important factor in overall compression performance, with a simple uniform prior over our DSL outperforming GZIP. Finally, we observe modest gains for adding inline execution feedback.

In §5.3, we conduct a thorough analysis of our results. We find that gains on synthetic data partially generalize to real data – models trained for longer perform better on short sequences, but all current models perform poorly on long sequences, suggesting KT can act as a useful test-bed to evaluate scaling properties of new methods. Finally, we perform an error analysis and find that models make a wide range of errors, from generating over-simplified programs that do not produce the input sequence, to repeating the input sequence in over-complicated ways.

To summarize, our main contributions are:[3]

- We propose the KOLMOGOROV-TEST, an extremely challenging compression-as-intelligence test for CODELMS (§3).
- We show that CODELMS can outperform previous compression methods on synthetic distributions where sampling program-sequence pairs is possible and efficient priors can be used, but fare very poorly on real data (§5).

---

[3]To support future progress, we release our code, data, and a public leaderboard.

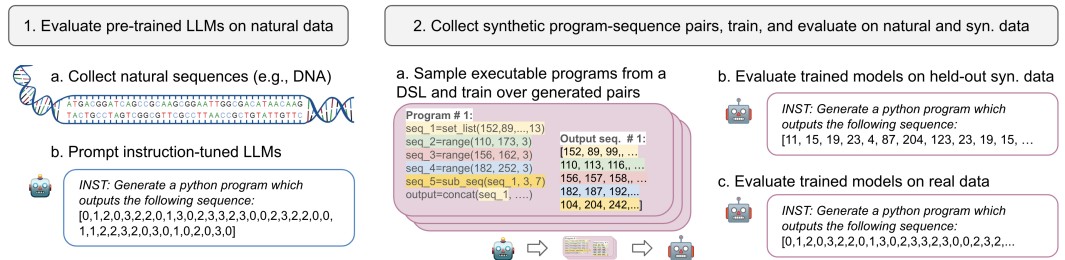

Figure 2: **Our main experimental settings.**

- We show that performance on real data scales poorly with increasing synthetic dataset size, suggesting that new breakthroughs may be needed for further progress on KT.

## 2 BACKGROUND

**Generative Modelling, Information Theory and Compression.** Generative modelling and compression are deeply related. Given a generative model over sequences (e.g., autoregressive transformer (Radford et al., 2018)) $p(x_i|x_{<i})$, one can use the arithmetic coding algorithm to compress a particular sequence $x$ to a bitstream of length about $-\sum_i \log_2 p(x_i|x_{<i})$ bits (Rissanen, 1976; Pasco, 1977). If the sequences are sampled from the gold distribution $p^*$, then the expected arithmetic code length is the cross entropy between $p^*$ and $p$. When $p = p^*$, the cross entropy equals the entropy, which is the fundamental limit on average compression length for data sampled from $p^*$.

Similarly, given a latent variable model $p(x, z)$, one can encode a sequence $x$ by first encoding a code $z$ using *prior* $p(z)$ and then encoding $x$ using likelihood $p(x|z)$. Using an optimal code for $p(z)$ and $p(x|z)$ the coding cost will be roughly $-\log p(z) - \log p(x|z)$. If we obtain $z$ by sampling from an encoder network $q(z|x)$, the expected coding cost will be $\mathbb{E}_q[-\log p(z) - \log p(x|z)]$ (Habibian et al., 2019), which (up to an additional entropy bonus $H(q)$) equals the evidence lower bound (ELBO) used for instance for VAE training (Kingma & Welling, 2022). Thus, both for autoregressive and latent variable models, maximizing likelihood is maximizing compression.

**Algorithmic Information Theory, Solomonoff Induction, and Kolmogorov Complexity.** In classical generative modelling and compression, one is concerned with finding a generative model from a particular model class that can be used to compress data from a particular distribution. By contrast, in algorithmic information theory, one considers as "model class" the set of all computable functions. This class trivially contains the optimal compression, and changing the computational model (programming language / universal Turing machine) only incurs a constant overhead depending on the pair of languages involved and regardless of the input (Grunwald & Vitanyi, 2008, 2.2.2).

In analogy to the discussion above, we can sketch the theory as follows, inspired by Solomonoff's theory of inductive inference (Solomonoff, 1964). We consider the *universal prior* $p(\rho) = 2^{-l(\rho)}$ over programs $\rho$ (where $l(\rho)$ denotes the length) that assigns higher probability to shorter programs ("Occam's Razor"). Furthermore, we consider the likelihood $p(x|\rho)$ over output sequences $x$ given $\rho$ which puts all mass on the actual output of $\rho$.

Then, treating $\rho$ as latent we may encode $x$ using a two-part code by finding a *program* $\rho$ that outputs $x$ (e.g. using a neural network $q(\rho|x)$), and encoding it using the *prior* $p(\rho)$. As before, the coding cost will be $\mathbb{E}_q[-\log p(\rho) - \log p(x|\rho)]$, where the second term is $\infty$ whenever $x$ is not the output of $\rho$, and 0 when it is. Clearly, the optimal $\rho$ is the shortest program that outputs $x$. The length of this program $\rho^*$ is called the *Kolmogorov Complexity $K(x)$* (Kolmogorov, 1963; Li & Vitányi, 1997).

## 3  DATA COLLECTION

### 3.1  PROBLEM SETTING

The theory presented above suggests an interesting challenge for CODELMs: to generate concise *programs* (under well-defined *priors*) that output a given sequence (Fig. 1 and Alg. 2 in §A.3). We evaluate CODELMs on three naturally occurring modalities: audio, text, and DNA (Fig. 2, left, and §3.2). We focus on audio and textual data, popular modalities in compression works with strong baselines (Deletang et al., 2024). We also introduce DNA sequences as these follow simple biological patterns. Although real data is useful for evaluation since it can be obtained in large amounts for many modalities, corresponding *programs* required for supervised training are not available. As a remedy, we experiment with synthetic settings where we collect program-sequence pairs by sampling programs from a Domain Specific Language (DSL) and executing the sampled programs (Fig. 2, right, and §3.3).

### 3.2  NATURALLY OCCURRING SEQUENCES

In all our experiments on natural data, the model has to compress 1MB of information. The length of the input sequence presented to our models ranges from 16 to 1024 bytes depending on the setting.

**Audio.**  We randomly sample audio snippets from the LibriSpeech development and test sets (Panayotov et al., 2015). We parse the data to three audio formats: (a) high quality audio with 16 bits depth, from which the original data can be perfectly recreated but each sample is split across two bytes, (b) lower quality audio with 8 bits depth, which is simpler for our baselines (§5), and (c) an estimation of the Mel-Frequency Cepstral Coefficients encoding (MFCC) which does not allow direct reconstruction but describes the main features of the sound (see §A.1 for more details). Each byte is represented as a number in range $[0, 255]$.

**Text.**  Following recent work (Deletang et al., 2024), we use the enwik9 Wikipedia corpus (Hutter, 2009). The Unicode text is encoded to a sequence of bytes with UTF-8 encoding which are represented as a list of numbers.

**DNA.**  We use Genome assembly GRCh38 which contains 3.1GB of human DNA in FASTA format (NCBI, 2023). Since the original files include upper and lower case variants of each of the four nucleotides, we use a vocabulary of eight characters represented by numbers in range $[0, 7]$[4].

### 3.3  SYNTHETIC DATA GENERATION

Our aim is to examine whether CODELMs can be trained to generate concise and correct programs. Hence, we adhere to the following desiderata for our data generation process: (a) **completeness**: each sequence has some probability for a program that produces it to be sampled, and (b) **simplicity**: shorter programs will be sampled with higher probability.

**Domain Specific Language.**  We design a *compositional* DSL, in the sense that the output sequence is created by *composing* simpler sub-sequences (Fig. 2, center). Our DSL supports the following four classes of functions (see §A.2 for a list of all supported functions):[5]

- **Sequence initiators**: functions that take variables as input and return a sequence of numbers. These include a range of increasing numbers, a repetition of a single number, or a fixed (hard-coded) list of numbers.
- **Sequence modifiers**: functions that take a sequence as input and return a modified version of the sequence, e.g., by reversing, repeating, or substituting elements from the sequence. A sub-class of our modifiers includes mathematical operations, such as scan-adding the elements in the sequence or applying a modulo operation.

---

[4]Lower case variants represent low-complexity or masked regions (e.g., repeats or predicted sequences), which we keep to allow perfect reconstruction of the input sequence.

[5]We note that for simplicity, our DSL does not allow recursion – it is a proper subset of primitive recursive functions and of Turing-computable functions (Turing, 1937) and thus deviates from the theory in §2.

Figure 3: **Two examples of program-sequence pairs from our synthetic data generation process.**

- **Sequence filters**: functions that filter a sequence, e.g., by keeping only even values.
- **Sequence mergers**: functions that take two sequences and merge them to a single one: concatenation, interleaving, pointwise addition, subtraction or modulo of the sequences.

**Sampling programs.** Given the compositional nature of our DSL, we sample programs similarly to the way sequences are sampled from a Context Free Grammar (Chomsky, 1956; Hopcroft et al., 2006). We can also control the distribution of programs by assigning priors over the program distribution, e.g., the lengths of the initiated sequence or probability to apply modifications. For simplicity and to avoid biases we keep the priors uniform[6]. For examples of program-sequence pairs, see Fig. 3. As in the natural data domains, we generate 1MB of evaluation data. For training, we sample 1M pairs from the same distribution. We provide additional details and statistics in §A.2.

**Encoding programs.** We encode programs written in our DSL using a factorized uniform prior over functions and arguments. As each line includes a call to a single function, the encoding cost of each line is the cost of encoding the function (i.e., $\log_2(|functions|)$) plus the cost of the input parameters (e.g., the number of bits needed to set an arbitrary list for *set list*, or the indices of the sequences in *concatenate*). Our code length calculation algorithm is presented in §A.2, Alg. 1.

# 4 EXPERIMENTAL SETUP

**Baselines.** For our classical compression baseline, we use GZIP (Deutsch, 1996). We also include a Language Modeling is Compression (LMIC) baseline (see §B.1 for implementation details), where the LM predictions are used together with arithmetic coding to compress the sequence as discussed in §2 (Deletang et al., 2024). A major limitation of LMIC is that the original LLM is needed to decompress the data[7]. We also include REPEAT, a naive baseline that always returns the input sequence, and an UPPER BOUND baseline that returns the matching program for our synthetic pairs.

**Zero-shot prompted baselines.** We prompt our models to generate the shortest Python program that produces the input sequence (see §B.1, Fig. 9 for the full prompt). We use strong open-weight models from the LLAMA-3.1 family with 8, 70, and 405 billion parameters Grattafiori et al. (2024). In addition, we use GPT4-O as a closed-source alternative (OpenAI et al., 2024b). We also experiment with a chain-of-thought (CoT) (Wei et al., 2022) baseline (see prompt in Fig. 10) and with OPENAI O1-MINI (OpenAI et al., 2024a), a closed-source model trained to think before generating the final answer. We use GZIP to encode programs before measuring their bit length (i.e., we use GZIP as a *prior* over programs) in all experiments, unless stated otherwise.

**Trained models.** For our trained models, we use LLAMA-3.1-8B as base model in addition to a 1.5-billion parameter LLM with the same architecture, which we train on a mixture of open-source code and text (see §B.1 for more details and technical specifications). We further train our models on 10K, 100K or 1M unique programs-sequence pairs sampled from our data generator (§3.3), and

---

[6]We note that this does not guarantee that each operator is used with the same probability, as some operators are only applicable in specific contexts (e.g., an addition between two sequences is only applicable when the sequences have the same length and the sum of the matching elements is in the allowed range).

[7]As our focus is on compressing relatively small amounts of data, we report raw LMIC results disregarding the weights of the model, an upper bound of the true performance which requires the weights that were used for encoding at decoding time.

name them SEQCODER-10K, SEQCODER-100K and SEQCODER(-1M), respectively. We use the encoding algorithm defined in §3.3 for encoding programs from our synthetic DSL.

**Sequence lengths.** We use a sequence length of 1024 bytes for our LMIC baseline, based on findings that further increasing context length does not result in significant gains (Deletang et al., 2024). For natural data, we use a sequence length of 128 bytes (see §5.3 for a discussion on input lengths). Our synthetic data has an average sequence length of 75.9 bytes with a standard deviation of 73.7 (see §A.2, Fig. 7 and Fig. 8 for histograms of sequence and programs lengths).

**Metrics.** Given a sequence $x$ and a program candidate $\rho$ suggested by the model with output $[\rho]$, we define the variable $y$ that backs off to the REPEAT baseline when $\rho$ errs by:

$$y = \begin{cases} \rho & \text{if } [\rho] = x, \\ x & \text{otherwise.} \end{cases}$$

We define the following metrics:

$$\text{Accuracy}(x, \rho) = \mathbf{1}_{\{[\rho] = x\}},$$
$$\text{CompressionRate}(x, \rho) = \frac{1 + \|y\|_{\text{enc}}}{\|x\|},$$
$$\text{Precision}(x, \rho) = \frac{\|\rho\|_{\text{enc}}}{\|x\|_{\text{gzip}}} \qquad \text{where } [\rho] = x,$$

where $\| \cdot \|$ denotes the bit-length of the data, $\| \cdot \|_{\text{gzip}}$ denotes the length after GZIP-encoding, and $\| \cdot \|_{\text{enc}}$ denotes the length after encoding programs with GZIP for prompted models and our uniform encoding for SEQCODER models. In other words, Accuracy measures whether the program is correct, CompressionRate measures the compression of the input when storing the compressed representation (an additional bit is saved to determine whether $y$ is $\rho$ or $x$ and lower Compression-Rate is better[8]), and Precision measures the compression of correct programs in relation to the GZIP baseline. REPEAT has an Accuracy, CompressionRate and Precision of 1 by definition.

## 5 RESULTS

Running prompted pretrained models as well as smaller trained models on KOLMOGOROV-TEST with natural as well as synthetic sequence data sources, we make the following main findings: Zero-shot compression by code generation is a challenging benchmark even for the most powerful models available (§5.1). On short sequences of synthetic data, trained specialized models outperform the in-context reasoning baseline LMIC as well as classical baseline algorithms (§5.2). In additional analyses, we examine the effect of scaling model and dataset size for models trained on synthetic data, their length generalization as well as typical failure modes (§5.3).

### 5.1 KOLMOGOROV-TEST IS CHALLENGING FOR CURRENT MODELS

Tab. 1 presents results for our prompted and SEQCODER models on different data modalities. Prompted models perform poorly – strong LLAMA-3.1-405B and GPT4-O models err on average more than 78% and 40% of the time for naturally occurring sequences, with high variance between modalities, and more than 65% and 55% of the time on our synthetic data, which follows simple compositional rules by design.

Nevertheless, larger models perform better, with LLAMA-3.1-405B reaching the best Accuracy between LLAMA models, and GPT4-O reaching the highest Accuracy and lowest Precision overall. All prompted models have a Precision score that is higher than 1.0, meaning that on average, generated programs are larger than the GZIP-encoded sequences. We partly attribute this to GZIP being unoptimized to the generated Python programs, a phenomenon we discuss in more detail in §5.3.

---

[8]CompressionRate is higher than 1 when the compressed representation is larger than the original.

| | Audio-16-bit | | Audio-8-bit | | Audio-MFCC | | DNA | | Text | | Syn. | |
| --- | --- | --- | --- | --- | --- | --- | --- | --- | --- | --- | --- | --- |
| | Acc. | Prec. | Acc. | Prec. | Acc. | Prec. | Acc. | Prec. | Acc. | Prec. | Acc. | Prec. |
| LLAMA-3.1-8B | 5.9 | 1.54 | 3.9 | 1.74 | 8.8 | 1.51 | 3.7 | 2.34 | 1.4 | 3.12 | 8.5 | 2.48 |
| LLAMA-3.1-70B | 18.0 | 1.96 | 10.1 | 1.67 | 24.2 | 1.56 | 22.5 | 2.18 | 9.6 | 3.17 | 18.0 | 2.78 |
| LLAMA-3.1-405B | 35.6 | 1.66 | 15.0 | 1.66 | 29.6 | 1.58 | 24.8 | 2.06 | 6.5 | 3.17 | 33.3 | 2.18 |
| GPT4-O | 69.5 | 1.34 | 36.4 | 1.43 | 83.8 | 1.33 | 50.3 | 1.73 | 54.2 | 1.94 | 44.7 | 1.65 |
| SEQCODER-1.5B | 0.0 | n/a | <0.1 | n/a | 0.1 | n/a | 0.0 | n/a | 0.0 | n/a | 84.5 | 0.57 |
| SEQCODER-8B | <0.1 | n/a | 0.1 | n/a | 0.1 | n/a | 0.0 | n/a | 0.0 | n/a | 92.5 | 0.56 |

Table 1: **Accuracy and Precision for zero-shot prompted models on the KOLMOGOROV-TEST for sequences of length** 128**.** Larger models have higher Accuracy. On synthetic distributions when sampling program-sequence pairs is possible, our trained SEQCODER performs best and is the only model to achieve Precision that is lower than 1.0, i.e., it outperforms GZIP when programs are correct. However, it performs poorly on real data.

Our trained SEQCODER models significantly outperform all CODELMs on synthetic data, and are the only models with a better Precision than the naive REPEAT baseline (which has a perfect Accuracy and a Precision of 1.0 by definition), suggesting that training data and strong priors are necessary for good performance. Nevertheless, it reaches near-zero performance on natural sequences. We discuss generalization to real data in more detail in §5.2 and §5.3.

We present results for LLAMA with CoT prompting in §B.2, Tab. 4. We observe similar trends, albeit lower performance for LLAMA-3.1-8B and LLAMA-3.1-70B (a drop of 1.3 and 5.1 points in Accuracy on average across modalities), and a small increase of 2.1 points in Accuracy for LLAMA-3.1-405B. For O1-MINI, we observe overall similar results to GPT4-O (see §B.2, Tab. 5 for mode details). Overall, our results are in line with recent work showing that CoT does not consistently improve performance for code generation tasks (Sprague et al., 2025; Zheng et al., 2025).

## 5.2 TRAINED CODELMs OUTPERFORM PREVIOUS COMPRESSION ALGORITHMS

Fig. 4 presents CompressionRate on synthetic data for SEQCODER models. Even though instruction-tuned models struggle on this distribution, models can be trained to outperform previous compression algorithms, reaching a CompressionRate of 0.38 for SEQCODER-8B, outperforming the strong GZIP and LMIC baselines *without requiring gigabytes of LLM weights* for decoding (in §B.2 we empirically show that GZIP and LMIC are strong baselines and that scaling also improves Accuracy on synthetic data).

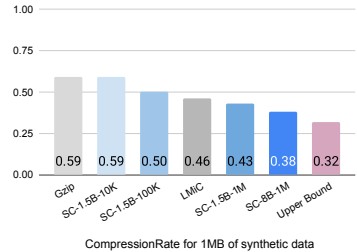

CompressionRate for 1MB of synthetic data

Figure 4: **CompressionRate for SEQCODER models trained on 10K-1M program-sequence pairs.** Models trained on enough data outperform the baselines.

**Length generalization.** To better test if improvements on synthetic data generalize to real data, we evaluate SEQCODER models on 10,000 shorter Audio-MFCC sequences of lengths 16-64, presented in Fig. 5. All models perform poorly on longer sequences of length 64. Models trained on 100K and 1M examples significantly outperform SEQCODER-1.5B-10K on shorter sequences, and larger SEQCODER-8B perform best.

Fig. 6 presents the length of correct prefixes for SEQCODER on Audio-MFCC sequences of length 128. SEQCODER-1.5B models trained on 10K examples fail early, i.e., more than 99% of programs have an error in the first 16 characters, even though the model correctly produces programs for 27.2% of sequences of the same length (Fig. 5). Models trained on 100K and 1M examples perform better, but they exhibit similar trends, and err in the first 16 characters in 93.5% and 88.7% of test cases. We provide additional evidence that longer synthetic programs and sequences are challenging for SEQCODER-1.5B in §B.3, Fig. 11a and Fig. 11b.

Tab. 2 presents results for GZIP and LLAMA-3.1-405B on a random sample of 2,000 Audio-8-bit sequences of lengths 16-128. As expected, CompressionRate for GZIP improves with length, as it is able to better exploit patterns in the input data (we extend to longer sequences for GZIP and LMIC in §B.2, Tab. 6, showing that these are strong baselines and CompressionRate is correlated

| Length | GZIP CR | LLAMA-405B Acc. | LLAMA-405B Prec. |
|---|---|---|---|
| 16 | 0.966 | **47.1** | 3.20 |
| 32 | 0.846 | 29.9 | 2.70 |
| 64 | 0.736 | 20.9 | 1.82 |
| 128 | 0.622 | 14.2 | **1.67** |
| 1M | **0.398** | n/a | n/a |

Table 2: **Effect of input length on Audio-8-bit sequences.** Performance improves for GZIP and CompressionRate as lengths increase, but Accuracy decreases.

|  | Syn. | Audio |
|---|---|---|
| SQ-1.5B | **86.6** | 6.5 |
| + feed only | 77.6 | 8.6 |
| + feed & ex | 75.9 | **8.7** |

Table 3: **Accuracy for synthetic sequences and Audio MFCC sequences of length 64 bytes with and without execution feedback.** Execution feedback is not sufficient to close the gap between synthetic and real data.

with length). However, for prompted models, longer sequences are more challenging and Accuracy decreases with length. Overall, these results suggest that while scaling to large models is beneficial, generalization to long sequences and from synthetic to real distributions remains a major challenge.

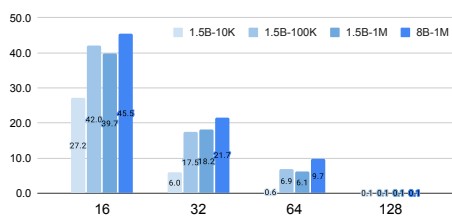

SeqCoder Accuracy on audio MFCC sequences of length 16-128

Figure 5: **Accuracy for our SEQCODER-1.5B on Audio MFCC sequences of lengths 16-128.** Models trained on more examples are significantly better on short sequences, but all models struggle on longer ones.

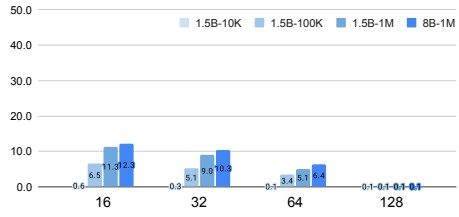

% of SeqCoder Programs with Correct Prefix of Length 16-128

Figure 6: **The longest correct prefix length for SEQCODER on Audio MFCC sequences of length 128.** Models trained on more examples and more parameters generate longer correct prefixes.

### 5.3 ADDITIONAL ANALYSES

**Effect of execution feedback.** To examine the effect of adding execution feedback (Yang et al., 2023; Shojaee et al., 2023; Ni et al., 2024), we experiment with two variants: (a) **feedback only during training**: execution feedback is added to training examples, and (b) **feedback & execution**: the code is executed by an interpreter at each line and results are presented to the model before the next step. In both cases the feedback is added as an end-of-line comment (see §B.3, Fig. 12).

Tab. 3 presents our results for 1,280 synthetic and 1,280 Audio-MFCC sequences of length 64. Execution feedback leads to small gains on real data, but also decreases performance on synthetic data and is not sufficient to significantly improve performance. This can be partially attributed to the fact that most training is spent on feedback tokens and the model is only trained on gold programs, and hence is not trained to dynamically fix model errors or bad trajectories (Ni et al., 2024).

**Repetitions of input data.** Python is highly expressive – there are many ways the model can repeat input sequences (e.g., by extracting to multiple sub-sequences that are later concatenated, see §B.3, Fig. 13 and Fig. 14). To check how often repetitions happen, we perform a qualitative analysis for 25 correct programs per modality (chosen randomly) for our LLAMA-3.1-405B and GPT4-O prompted models, presented in Tab. 9. We find that repetitions account for most correct predictions on natural data (more than 95% on average), but only 20% on our synthetic sequences, where models are able to leverage the synthetically induced patterns (Fig. 15). Hence, the low

Precision of our prompted models can be partly attributed to the GZIP prior being under-optimized to Python programs.

**Error analysis.** To understand the errors our models make, we manually analyze 25 errors per modality for LLAMA-3.1-405B and GPT4-O, presented in Tab. 10. For both natural and synthetic sequences, most errors are caused by models misusing specific operators during execution, e.g., the wrong number of repetitions for a character (see Fig. 16 and Fig. 17 in §B.3). This can be attributed to the fact that current LLMs still struggle with symbolic reasoning and counting in particular (Hendrycks et al., 2021; Yehudai et al., 2024; Ball et al., 2024).

Additionally, we identify a couple of specific failure cases. In more than 30% of cases, models try repeating the input sequence by breaking it to sub-sequences that are later merged, but still introduce subtle errors when constructing and merging these sub-sequences (Fig. 18). We identify a special failure case for the text modality, where models successfully reverse-engineer our data generation process and generate a string which they convert to UTF-8-encoded bytes, but often with subtle errors leading to low success rates (Fig. 19).

# 6 RELATED WORK

**Data compression.** Data compression is one of the main problems in computer science & communications. In addition to the works described in §2, fundamental work includes Shannon (1948); Cover & Thomas (2006). See Salomon (2004); Lelewer & Hirschberg (1987); Jayasankar et al. (2021) for detailed surveys. A major line of work is devoted towards domain-specific algorithms, including specifically for text, audio, and DNA (Storer & Szymanski, 1982; Neuburger, 2010; Salau et al., 2019; Du et al., 2020). Recently, it has been proposed that LLMs can be harnessed for compression via arithmetic coding (Valmeekam et al., 2023; Deletang et al., 2024). While the raw compression rate (excluding the weights of the LLM) of this method outperforms classical compression algorithms, the full LLM parameters are required for decoding. Our work differs by prompting and training CODELMs to directly generate *programs* that produce the original data.

**Evaluation of CODELMs.** Recent jumps in the coding ability of LLMs (Chen et al., 2021; Rozière et al., 2024) paved the path to benchmarks that focus on code generation. For example, Chen et al. (2021) on generating code given a function's signature, Austin et al. (2021) on solving coding interview questions, and Li et al. (2022) focus on competitive programming tasks. Other benchmarks focus on agentic settings where models interact with an environment to solve GitHub issues or run research experiments (Jimenez et al., 2024; Bogin et al., 2024). KT has several advantages over previous benchmarks: there is an abundance of diverse natural data, strong baselines for compression already exist, and evaluation metrics are well-defined. Our work suggests that making progress on KT is challenging and new innovations are needed for further progress. Moreover, to outperform current compression methods, CODELMs must model complex environments, potentially improving their ability to act as world models (Tang et al., 2024).

**Synthetic data augmentation for coding and symbolic reasoning.** Synthetic data augmentation has been recently shown to be highly effective in injecting numerical skills to LMs (Geva et al., 2020; Yoran et al., 2022; Lu et al., 2024; Mitra et al., 2024), however improvements do not necessarily generalize (Liu et al., 2023a). More complex pipelines for synthetic data generation with LLMs are often used for CODELMs (Rozière et al., 2024).

# 7 DISCUSSION AND LIMITATIONS

**Generalization from synthetic to real data.** Our results show that generalization from synthetic to real distributions remains a challenge. Future work can experiment with methods that tilt the synthetic distribution towards the real one, e.g., by filtering easy-to-detect synthetic sequences (Ganin & Lempitsky, 2015). An exciting direction for future research is to model KT as a reinforcement learning task (François-Lavet et al., 2018; Mnih et al., 2013) and learn from a reward when correct solutions are generated, with a bias towards shorter solutions (Ellis et al., 2023). Additionally, future methods can benefit from curriculum-learning where models learn simpler examples before more complex ones (Hacohen & Weinshall, 2019; Soviany et al., 2022).

**Additional modalities.** This work focuses on natural text, audio, and DNA sequences in addition to our synthetic data, and can be easily extended to cover additional modalities such as publicly-available image and video sequences (Deng et al., 2009; Soomro et al., 2012). An exciting direction for future work is to mix training examples from different domains, composing of different patterns and functionalities. Additionally, we only focus on the ability of CODELMS to compress sequences of data, and leave compression of executable programs for future work.

**Compressing larger amounts of data.** In this work, we focus on compressing 1MB of data, a smaller amount than previous work (Deletang et al., 2024) and the Hutter Prize (Hutter, 2009), which focus on 1GB of data per modality. We opt for 1MB sequences as this amounts to roughly 60K sequences across all modalities (around 20K for 1MB of DNA and 7800 sequences for each of the other modalities), a relatively large amount of examples in terms of current code generation benchmarks (Chen et al., 2021; Austin et al., 2021; Li et al., 2022; 2023). Nevertheless, compressing larger sequences is likely to be feasible as CODELMS and hardware continue to improve. To assist further progress, we will also release 1GB versions of our datasets.

**Scaling to longer sequences.** In our synthetic experiments, we set the maximum size of a sub-sequence to 25, and the maximum number of initiated sub-sequences to 5 (see §A.2 for all hyper-parameters). Hence, a program will not produce a sequence of more than 125 random characters (concatenating 5 sub-sequences of 25 random characters), but can produce longer sequences by using *modifiers*, e.g., by repeating a sub-sequence. Extending our DSL to longer sequences, for example by further scaling training compute, can be an interesting direction for future research.

Moreover, longer sequences sequences yield lower compression rates, as exemplified for GZIP in Tab. 2. Although we mainly experiment with sequences of length 128 due to the low Accuracy of current models on long sequences, future models might be able to achieve better compression on much longer sequences.

**Other languages** We have experimented with general Python as well as a simple sequence-oriented DSL. Future work could explore how the choice of language affects the performance of models, the rate of training progress, and the degree of generalization to OOD sequences.

**Stronger priors.** In our experiments we used the GZIP prior for programs from our prompted models and our uniform prior over our synthetic DSL. Future work can experiment with stronger priors, e.g., small LMs that are trained on the distribution of programs. Another possible future direction is to focus on lower level languages where developing strong priors might be simpler. Although such programs might be harder for humans to understand, the potential of CODELMS to generate novel solutions is an exciting opportunity for future work.

**Runtime compute.** In our experiments, we do not consider the runtime compute of programs execution. To limit the effect of inefficient programs, we limit the runtime of each program to five seconds and consider programs that run for longer as *non-executable* (similar to runtime exceptions). We provide additional statistics on failed and non-executable programs in §B.2. Similarly, we do not consider the weights of the LLM (which are only required for generating the programs) or the cost of the Python standard library required to execute the programs. Additionally, we leave experimenting with specialized tokenization methods, e.g., Dagan et al. (2024), for future work.

## 8    CONCLUSION

In this work we introduce the KOLMOGOROV-TEST, a compression-as-intelligence test for code generation models, based on the notion of *Kolmogorov complexity*. To evaluate current models, we use real audio, text, and DNA data, and develop an automatic framework for sampling program-sequence pairs. KT is extremely challenging for current models, including LLAMA-3.1-405B and GPT4-O. Additionally, for synthetic distributions where pairs can be automatically sampled, trained models have a better CompressionRate than current methods. As our method can potentially be scaled to an infinite number of real problems of varying difficulty, it can provide a challenging evaluation test-bed for future code generation models and pave the path towards new methods that compress large amounts of real data.

## REPRODUCIBILITY

We have made significant efforts to ensure our work can be reproduced. Our datasets will be fully released, including our DSL and synthetic data generation framework. We will also release larger 1GB variants to facilitate future research. In addition, we will release code to reproduce results with our prompted models and will host a live leaderboard to allow the community to monitor future developments. As we will not be able to open-weight our SEQCODER models, we will release training code for SEQCODER-8B models on our synthetic pairs, enabling reproduction of our results and simplify training new SEQCODER models.

## ETHICS STATEMENT

In this work, we introduce the KOLMOGOROV-TEST, a challenging intelligence test for CODELMs based on the notion of Kolmogorov complexity. While we do not see any immediate risk caused by KT, code generation research is not without risk. Powerful models can be used for malicious reasons, e.g., by hackers or other malicious users. If compression via zero-shot code generation was solved, it might enable a leap in telecommunications efficiency while at the same time raising concerns about opaque behavior of strong artificial intelligence.

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

## A  DATA GENERATION

**Additional details and statistics.**   We randomly choose audio sequences from LibriSpeech until reaching 1MB of data. As we opt not to trim sequences, we are left with 1,015,360 bytes in the 16-bit setting, 1,00,3280 bytes in the 8-bit setting, and 1,000,960 bytes in the MFCC setting. Similarly, we have 1,004,232 text bytes, and 1,000,105 synthetic bytes. For DNA, we have exactly 1MB.

The maximum sequence length shown to our models is 128. As we do not merge between different origin sequences (i.e., an audio file or synthetic sequence), some sequences can be shorter. The average sequence length for natural sequences is 75.9 for our synthetic data, 126.0 for MFCC and 127.9 for all other modalities. We provide detailed statistics regarding synthetic program-sequence lenghts in §A.2.

**Figures.**   The DNA illustration in Fig. 2 is courtesy of The National Human Genome Research Institute[9].

### A.1  NATURALLY OCCURRING SEQUENCES

**Audio MFCC.**   We use Librosa[10] to parse our audio files to MFCC. Specifically, we parse each file to 20 MFCCs. As MFCCs are floats, we round each float into an integer which we take modulo 256 to obtain numbers in the range $[0, 255]$. Note that this representation is far from being reconstructible but we posit that it provides a source of richly structured natural data nevertheless.

### A.2  SYNTHETIC DATA GENERATION

**Automatic generation of program-sequence pairs.**   Our DSL supports the following functions and can be easily extended in the future:

- **[Initiator]** *Set list*: Initiates a fixed sequence of numbers.
- **[Initiator]** *Range*: Initiates a sequence of increasing numbers from start to end index, either with or without a step interval.
- **[Initiator]** *Repeat*: Initiates a sequence of repeated occurrences of the same number.
- **[Modifier]** *Substitute*: Substitutes a value in a list with a different value.
- **[Modifier]** *Reverse*: Reverses a sequence.
- **[Modifier]** *Sub-sequence between indexes*: Returns a sub-sequence between two indexes, either with or without intervals.
- **[Modifier]** *Repeat list*: Returns a repetition of a sequence a specified number of times.
- **[Modifier]** *Max/Min*: Returns the $n$ maximum or minimum items in the sequence.
- **[Modifier]** *Add/Subtract/Modulo*: Applies a mathematical operation (addition, subtraction, or modulo) with a specified fixed operand to all elements in the sequence.
- **[Modifier]** *Scan add*: Returns a new sequence where the value at index $n$ is the sum of all items up to that index.
- **[Filter]** *Is even/odd*: Keeps only even or odd numbers in the sequence.
- **[Filter]** *Is non-zero*: Keeps only non-zero numbers in the sequence.
- **[Merger]** *Add/Subtract/Modulo two lists*: Applies a mathematical operation (addition, subtraction, or modulo) over elements at matching indexes in two sequences.
- **[Merger]** *Concatenate*: Concatenates two sequences.
- **[Merger]** *Interleave*: Interleaves one sequence with another in alternating indexes. If one sequence is longer, the remaining values will form the suffix of the interleaved sequence.

---

[9]https://www.genome.gov/genetics-glossary/acgt
[10]https://librosa.org/doc/main/generated/librosa.feature.mfcc.html

When sampling from our DSL, we first sample the initiators, then produce additional sequences using sampled filters and modifiers before the sub-sequences are composed by sampled mergers to form the final sequence. We can control the distribution of programs via the hyper-parameters of the sampling process.

The number of initiators is uniformly sampled in range $\{1, ..., 5\}$, the length of each fixed sequence is sampled uniformly in range $\{5, ..., 25\}$.

The probability to apply any non-mathematical modifier (e.g., *repeat* or *substitute*) is set to $0.4$, and the probability for each specific modifier is set to $0.1$. After the modification, we reuse the original sequence at probability $0.2$. For substitution, we exclude the original sequence at probability $0.25$. Similarly, the probability to apply any mathematical modifier (e.g., *Add/Subtract/Modulo*) to an initiated sequence or filter is set to $0.4$, and the probability for a specific filter is set to $0.1$. Mathematical mergers are treated similarly – the probability to any mathematical merger is $0.4$ and for a specific merger is $0.1$. Finally, the probability for concatenate and interleave operations is set to $0.8$ and $0.2$, respectively, and we continue to concatenate and interleave sub-sequence until all sub-sequences are used in the final output sequence.

Additionally, we make sure that sequences in our evaluation data will not be part of a program-sequence pair during training. In case a sequence is generated by more than one program, we only keep the shortest program that produces the sequence in our dataset.

**Program-sequence lengths.** Fig. 7 and Fig. 8 presents the histogram of the lengths of our synthetic sequences and programs respectively.

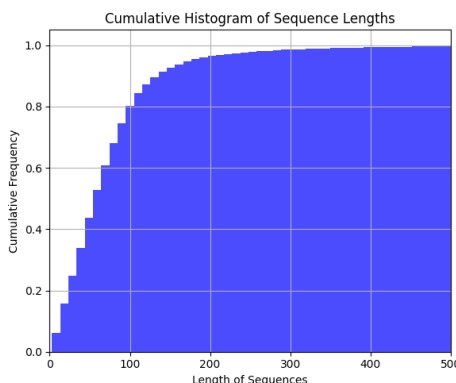

Figure 7: Cumulative histogram of sequence lengths for our synthetic distribution.

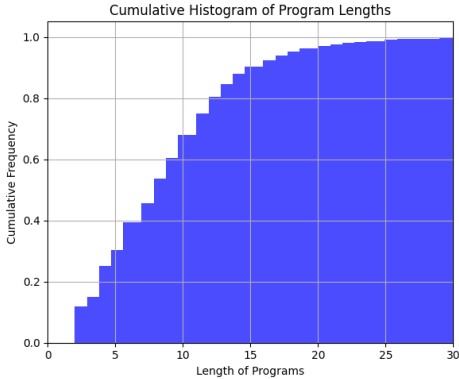

Figure 8: Cumulative histogram of program lengths (number of lines) for our synthetic distribution.

**Uniform prior over DSL programs.** Alg. 1 presents the number of bits needed to sample a program using the uniform prior over programs and arithmetic coding (Rissanen, 1976; Pasco, 1977). The cost to encode each line has two main components.

The first, is to encode the method that is being called. Since we use a uniform prior, this is always $log$ of the number of functions (stopping the generation and assigning one of the previously defined sequences as the output is formulized as an additional function).

The second, is to encode the variables that are being passed to the function, which is determined by the environment and is specific for each function. For example, setting a random list of elements (*set list*), requires $l * b$ bytes, $l$ indicating the length of the list, and $b$ indicating the number of bits in each byte. Similarly, concatenation two previously defined sub-sequences requires encoding the indexes of these sequences (which is a $log$ operation of the current number of lines), and repeating a character requires encoding the character (again, the byte size) and the number of repetitions ($log$ of the maximal number of allowed repetitions in the environment).

---

**Algorithm 1** Encoding a program under the uniform prior.

---

**function** ENCODE_FUNC_PARAMS(func, func_params, hps)
    mapping = { "set_list": $len(func\_params.list\_size) \times hps.byte\_size$, "concatenate": $\log_2(hps.line\_index) \times 2$, "repeat_num: $hps.byte\_size + hps.max\_num\_repetitions$, ... }
    **return** mapping[func]
**end function**
$x \leftarrow 0$
$bits\_per\_func \leftarrow \log_2(num\_functions + 1)$
**for all** $i, (f, f\_params)$ in enumerate(program_lines) **do**
    $x \mathrel{+}= bits\_per\_func$               ▷ # bits to encode the function type
    $x \mathrel{+}= encode\_func\_params(f, f\_params, hps, i)$    ▷ # bits to encode the function params
**end for**
**return** $x$

---

### A.3 FORMAL DEFINITION OF THE KOLMOGOROV-TEST

Alg. 2 presents a formal definition of the KOLMOGOROV-TEST. CODELMs that generate shorter programs have better compression rates (see §4 for our metrics and experimental settings).

---

**Algorithm 2** Compression by Code Generation (The KOLMOGOROV-TEST)

---

**Require:** Sequence $x$, CODELM $M$, Programs prior $p$
 1: Generate program $\rho$ with model $M$ that produces the sequence $x$
 2: Encode $\rho$ with prior $p$
 3: **return** $p(\rho)$

---

## B EXPERIMENTS

### B.1 MODELS

**GZIP baseline.** We use the python implementation of GZIP[11]. To enable reconstruction of the sequential data, we use new line as the separator between sequences.

**LMIC baseline.** We use our SEQCODER-1.5B model for the results reported in §5 and provide additional results in §B.2.

**Prompted baselines.** Fig. 9 presents our full prompt used in our prompted models experiments. We use vLLM[12] to set-up inference servers with the instruction-tuned LLAMA-3.1 models available on HuggingFace[13].

**Trained models.** Our SEQCODER-1.5B model is a transformer (Vaswani et al., 2017) with 1.5 billion parameters. The model follows a similar architecture to the recent LLAMA models (Touvron et al., 2023; Grattafiori et al., 2024) but with less parameters – it has 24 layers, a hidden-dimension of 2,048, multi-head attention with 16 heads, and a maximum sequence length of 2,048 tokens. To endow the model with coding and reasoning skills before training on synthetic data from our DSL, we pre-train on 200B tokens of publicly-available text and code. Although this model is under-trained in relation to current state-of-the-art models and hence significantly weaker, it achieves non-negligible performance on popular coding and reasoning tasks – Pass@1 of 11.6 on HumanEvalPlus (Liu et al., 2023b), and Accuracy of 28.2% on ARC (Clark et al., 2018) and 68.2% on PIQA (Bisk et al., 2020).

---

[11]https://docs.python.org/3/library/gzip.html
[12]https://github.com/vllm-project/vllm
[13]https://huggingface.co/meta-llama/Meta-Llama-3.1-{8B/70B/405B}

**System Prompt**

```
Generate a Python program that, when executed, reproduces a
specified input sequence.  The program should be as concise as
possible.
```

**Instructions**

```
Instructions:
- Write a multi-line Python program.  Each line should either
assign a new variable or define a new function.
- These variables and functions can be reused throughout the
program.
- Identify and utilize patterns in the input sequence to
minimize the length of the program.
- Assign the final output of the sequence to the variable
output.  This output will be used to verify the correctness
of the program.
- Do not include print statements or return statements.
- Ensure that the generated code is executable in a Python
interpreter without modifications.
- Do not include the python code block syntax in your
response.
- End your response with ###.

### Input Sequence:
#SEQ#

### Expected Output:
The Python program that generates the sequence is:
```

Figure 9: Full prompt used in our zero-shot experiments.

| | Audio-16-bit | | Audio-8-bit | | Audio-MFCC | | DNA | | Text | | Syn. | |
| --- | --- | --- | --- | --- | --- | --- | --- | --- | --- | --- | --- | --- |
| | Acc. | Prec. | Acc. | Prec. | Acc. | Prec. | Acc. | Prec. | Acc. | Prec. | Acc. | Prec. |
| LLAMA-3.1-8B | 2.9 | 1.6 | 1.60 | 2.05 | 5.4 | 1.59 | 6.9 | 3.8 | 0.4 | 2.67 | 6.9 | 3.8 |
| LLAMA-3.1-70B | 14.3 | 1.61 | 4.2 | 1.74 | 23.6 | 1.50 | 9.2 | 2.25 | 3.7 | 2.69 | 16.9 | 2.93 |
| LLAMA-3.1-405B | **37.9** | **1.53** | **10.4** | **1.60** | **49.7** | **1.41** | **23.4** | **1.94** | **5.1** | **2.35** | **30.5** | **2.6** |

Table 4: Accuracy and Precision for zero-shot prompted models on the KOLMOGOROV-TEST for sequences of length 128 with CoT. Trends are similar to those without CoT in Tab. 1.

## B.2 RESULTS

**Chain-of-thought prompting does not lead to additional improvements.** We present our CoT prompt in Fig. 10 and our experiments with CoT with LLAMA in Tab.4. Similarly to recent work Sprague et al. (2025); Zheng et al. (2025), we do not observe significant gains with CoT prompting.

We present the percentage of executable programs, accuracy for executable programs, and Accuracy for GPT4-O and O1-MINI in Tab. 5. While O1-MINI has a higher accuracy for executable programs (69.8 vs 63.2), it also has a lower percentage of executable programs (81.7 vs 89.9), and both models have similar Accuracy (57.2 vs 56.5), suggesting that while longer chains-of-thoughts are able to increase performance, they do not yet lead to major improvements on the KOLMOGOROV-TEST.

**GZIP and LMIC are strong baselines.** Tab. 6 presents results for our GZIP and LMIC baselines, for sequences of length 128 and 1,024[14]. For GZIP we also compress all 1MB as a single

---

[14]Each element in the sequence is a single token, and we use a comma separator between elements. We do not consider the cost of generating the separator token for the LMIC baseline.

**System Prompt**

```
Generate a Python program that, when executed, reproduces a
specified input sequence.  The program should be as concise as
possible.
```

**Instructions**

```
Instructions:
- Write a multi-line Python program.  Each line should either
assign a new variable or define a new function.
- These variables and functions can be reused throughout the
program.
- Identify and utilize patterns in the input sequence to
minimize the length of the program.
- Assign the final output of the sequence to the variable
output.  This output will be used to verify the correctness
of the program.
- Do not include print statements or return statements.
- Ensure that the generated code is executable in a Python
interpreter without modifications.
- Do not include the python code block syntax in your
response.
- End your response with ###.
- Before the program, you can use the Thought field to
generate how you think the task should be solved.  After
the thought, generate "The Python program that generates the
sequence is:", followed by the program.

### Input Sequence:
#SEQ#

### Expected Output:
The Python program that generates the sequence is:
```

Figure 10: Full prompt used in our zero-shot CoT experiments. The difference from our non-CoT prompt in Fig. 9 is highlighted.

| | GPT4-O | | | O1-MINI | | |
| --- | --- | --- | --- | --- | --- | --- |
| | % Ex. | Ex. Acc. | Acc. | % Ex. | Ex. Acc. | Acc. |
| Audio-16-bit | 88.2 | 78.8 | 69.5 | 81.2 | 87.7 | 71.2 |
| Audio-8-bit | 94.6 | 38.5 | 36.4 | 82.2 | 51.1 | 42.0 |
| Audio-MFCC | 94.4 | 88.8 | 83.8 | 86.8 | 93.8 | 81.4 |
| DNA | 93.1 | 54.0 | 50.3 | 85.6 | 67.1 | 57.4 |
| Text | 76.5 | 70.8 | 54.2 | 82.8 | 53.6 | 44.4 |
| Synthetic | 92.9 | 48.1 | 44.7 | 71.4 | 65.8 | 47.0 |
| Average | 89.9 | 63.2 | 56.5 | 81.7 | 69.8 | 57.2 |

Table 5: The percentage of executable programs (% Ex.), accuracy for executable programs (Ex. Acc.), and Accuracy (Acc.) for GPT4-O and O1-MINI. To reduce costs, O1-MINI was evaluated on 500 random sub-sequences of length 128. On average, both models have similar Accuracy. O1-MINI has higher accuracy for executable programs, but a lower percentage of executable programs.

sequence. We note that stronger compression baselines exist, and that the current state-of-the-art CompressionRate for our text data from the Hutter Prize is around 0.11 (Hutter, 2009).

LMIC is a strong baseline, it outperforms GZIP (with 1MB) on $5/6$ modalities with a sequence length of 1,024 and $4/6$ modalities with a sequence length of 128. Our results on Audio-8-bit are

| Sequence length | LMɪC | | Gzɪᴘ | | |
|---|---|---|---|---|---|
| | 128 | 1024 | 128 | 1024 | 1MB |
| Audio-16-bit | 0.697 | **0.639** | 1.147 | 1.014 | 0.920 |
| Audio-8-bits | 0.319 | **0.273** | 0.621 | 0.454 | 0.398 |
| Audio MFCC | 0.739 | **0.698** | 1.250 | 0.974 | 0.903 |
| DNA | 0.750 | **0.662** | 1.182 | 0.894 | 0.714 |
| Text | 0.611 | 0.463 | 0.782 | 0.525 | **0.357** |
| Synthetic | **0.449** | 0.450 | 0.943 | 0.819 | 0.593 |
| Average | 0.594 | **0.531** | 0.988 | 0.748 | 0.647 |

Table 6: CompressionRate for Gzɪᴘ and LMɪC with various input sequence lengths.

comparable to the Gzɪᴘ and Cʜɪɴᴄʜɪʟʟᴀ-1B baseline from Deletang et al. (2024) (which reports compression rates of 0.249 with LMɪC and 0.364 for Gzɪᴘ with 1GB of data, both within a a 3.6-point margin). On on text, our LMɪC baseline is significantly weaker than the one in Deletang et al. (2024). We attribute the difference to the different setting – we present the sequences as *decimal numbers* (to enable model to identify mathematical patterns), while the original work presents sequences as *ASCII characters*. Moreover, Gzɪᴘ improves as we increase the input length, reaching CompressionRate of 0.32 for 1GB of data, similar to Deletang et al. (2024).

**Scaling SᴇǫCᴏᴅᴇʀ models improves Accuracy on synthetic data.** Fig. 7 presents the Accuracy for the different SᴇǫCᴏᴅᴇʀ models on our held-out synthetic dataset. SᴇǫCᴏᴅᴇʀ-1.5B and SᴇǫCᴏᴅᴇʀ-8B models are trained for $20K$ and $10K$ steps, respectively. SᴇǫCᴏᴅᴇʀ-1.5B-10K overfits and accuracy decreases with additional training. SᴇǫCᴏᴅᴇʀ-1.5B (1M examples) significantly outperforms SᴇǫCᴏᴅᴇʀ-1.5B-100K and the larger SᴇǫCᴏᴅᴇʀ-8B performs better and learns faster.

**Scaling prompted models increases percentage of executable programs.** Fig. 8 shows the percentage of programs that execute without error and terminate within five seconds, broken down by model and dataset. Larger models generate more executable programs. Programs generated by Lʟᴀᴍᴀ-3.1-405B and GPT4-ᴏ are executable 77.2% and 89.9% of the time on average across modalities.

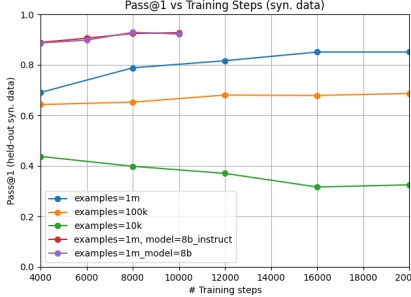

Table 7: Accuracy on our synthetic held-out set for the different SᴇǫCᴏᴅᴇʀ models.

| Modality | Lʟᴀᴍᴀ-3.1 | | | GPT 4-ᴏ |
|---|---|---|---|---|
| | 8B | 70B | 405B | |
| Audio-16-bit | 25.6 | 67.2 | 69.3 | **88.2** |
| Audio-8-bit | 28.2 | 82.0 | 87.4 | **94.6** |
| Audio-MFCC | 25.9 | 71.0 | 76.6 | **94.4** |
| DNA | 33.7 | **93.9** | 91.9 | 93.1 |
| Text | 4.3 | 51.0 | 49.1 | **76.5** |
| Synthetic | 42.0 | 85.4 | 88.9 | **92.9** |
| Average | 26.6 | 75.1 | 77.2 | **89.9** |

Table 8: The percentage of executable programs for our prompted models.

**Strong priors are necessary for good compression.** In §5, Fig. 4 we show that we can outperform previous compression methods for our synthetic data using SᴇǫCᴏᴅᴇʀ-1.5B and a uniform prior over programs and arithmetic coding. The same programs achieve a compression rate of 5.26 and 0.59 with raw (the size of the programs saved as strings on the disk) and Gzɪᴘ encoding, showing that strong priors are needed for good CompressionRate.

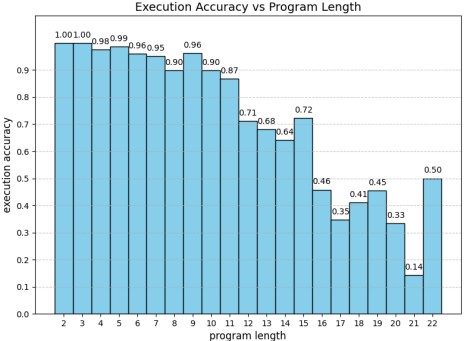

(a) SEQCODER-1.5B accuracy as a function of gold program lengths.

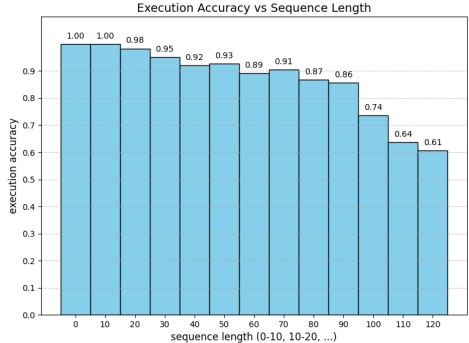

(b) SEQCODER-1.5B accuracy as a function of input sequence lengths.

Figure 11: SEQCODER-1.5B accuracy as a function of gold program and input sequence lengths.

---

**Model Input**

```
Generate a program whose execution results in the following sequence:  [18, 149,
19, 150, 20, 151, 21, 152, 22, 153, 23, 154, 24, 155, 25, 156, 26, 157, 27, 158,
28, 159, 237, 160, 237, 161, 237, 162, 237, 163, 237, 164, 237, 165, 237, 166,
237, 167, 237, 168, 237, 169, 237, 170, 237, 171, 237, 237, 237, 237, 237, 237,
237, 237, 237, 237, 142, 143, 144, 145, 146, 147, 148, 149, 150, 151, 152, 153,
154, 155, 28, 27, 26, 25, 24, 23, 22, 21, 20, 19, 18, 149, 150, 151, 152, 153,
154, 155, 156, 157, 158, 159, 160, 161, 162, 163, 164, 174, 166, 167, 168, 169,
170, 171, 18, 19, 20, 21, 22, 23, 24, 25, 26, 27, 177, 28, 177, 26, 25, 24, 23,
22, 21, 20, 19, 18, 28, 177, 26, 25, 24, 23, 22, 21, 20, 19, 18, 149, 150, 151,
152, 153, 154, 155, 156, 157, 158, 159, 160, 161, 162, 163, 164, 165, 166, 167,
168, 169, 170, 171]
```

**Gold Output (with in-line execution feedback)**

```
sequence_1 = range_func_up(149, 171)   # [149, 150, ..., 170, 171]
sequence_2 = range_func_up(18, 28)      # [18, 19, ..., 27, 28]
sequence_3 = repeat_num(22, 237)        # [237, 237, ..., 237, 237]
sequence_4 = range_func_up(142, 155)   # [142, 143, ..., 154, 155]
sequence_5 = reverse_list(sequence_2)  # [28, 27, ..., 19, 18]
sequence_6 = substitute(sequence_1, 165, 174) # [149, 150, ..., 170, 171]
sequence_7 = substitute(sequence_2, 28, 177)   # [18, 19, ..., 26, 177]
sequence_8 = substitute(sequence_5, 27, 177)   # [28, 177, ..., 19, 18]
sequence_9 = concatenate(sequence_2, sequence_3) # [18, 19, ..., 237, 237]
sequence_10 = concatenate(sequence_9, sequence_4) # [18, 19, ..., 154, 155]
sequence_11 = concatenate(sequence_10, sequence_5) # [18, 19, ..., 19, 18]
sequence_12 = concatenate(sequence_11, sequence_6) # [18, 19, ..., 170, 171]
sequence_13 = concatenate(sequence_12, sequence_7) # [18, 19, ..., 26, 177]
sequence_14 = concatenate(sequence_13, sequence_8) # [18, 19, ..., 19, 18]
sequence_15 = interleave(sequence_14, sequence_1)  # [18, 149, ..., 170, 171]
sequence_16 = concatenate(sequence_15, sequence_8) # [18, 149, ..., 19, 18]
sequence_17 = concatenate(sequence_16, sequence_1) # [18, 149, ..., 170, 171]
output = sequence_17 # [18, 149, ..., 170, 171]
```

Figure 12: An example program-sequence pair with in-line execution feedback.

## B.3 ANALYSIS

Fig. 11a and Fig. 11b present accuracy for SEQCODER-1.5B as a function of program and sequence length, respectively. Accuracy decreases with length of programs or sequences.

Fig. 12 presents an example for a synthetic program-sequence pair with in-line execution feedback.

Tab. 9 presents the percentage of repetitions in correct programs for LLAMA-3.1-405B and GPT4-O. Fig. 13 presents an example where LLAMA-3.1-405B succeeds by repeating the input sequence. Fig. 14 presents an example where LLAMA-3.1-405B repeats the input sequence in a complex

| | Audio-16-bit | | Audio-8-bit | | Audio-MFCC | | DNA | | Text | | Synthetic | |
|---|---|---|---|---|---|---|---|---|---|---|---|---|
| | Rep | No Rep | Rep | No Rep | Rep | No Rep | Rep | No Rep | Rep | No Rep | Rep | No Rep |
| LLAMA-3.1-405B | 84 | 16 | 96 | 4 | 100 | 4 | 100 | 0 | 96 | 4 | 20 | 80 |
| GPT4-O | 96 | 4 | 96 | 4 | 100 | 0 | 100 | 0 | 100 | 0 | 20 | 80 |

Table 9: The percentage of repetitions in correct programs from a qualitative analysis of 25 programs per modality for LLAMA-3.1-405B and GPT4-O. Most correct examples for natural sequences are because of repetitions, but the contrary is true for synthetic sequences.

| | 16-bit | | | 8-bit | | | MFCC | | | DNA | | | Text | | | Synthetic | | |
|---|---|---|---|---|---|---|---|---|---|---|---|---|---|---|---|---|---|---|
| | Ex | Re | Un | Ex | Re | Un | Ex | Re | Un | Ex | Re | Un | Ex | Re | Un | Ex | Re | Un |
| LMA-405B | 80 | 20 | 0 | 96 | 4 | 0 | 72 | 28 | 0 | 40 | 60 | 0 | 24 | 56 | 20 | 92 | 8 | 0 |
| GPT4-O | 72 | 28 | 0 | 88 | 12 | 0 | 40 | 60 | 0 | 36 | 64 | 0 | 44 | 48 | 8 | 96 | 4 | 0 |

Table 10: The percentage of execution (Ex), repetition (Re), and Unicode (Un) errors in erroneous programs from a qualitative analysis of 25 programs per modality for LLAMA-3.1-405B and GPT4-O. More than a third of the errors for natural sequences are caused by repetition errors.

and verbose manner, which is sub-optimal for the GZIP prior. Fig. 15 presents an example where LLAMA-3.1-405B succeeds on a synthetic sequence by leveraging the induced patterns.

Tab. 10 presents our qualitative error analysis on 25 erroneous programs per modality for LLAMA-3.1-405B and GPT4-O. Fig. 16, Fig. 17, Fig. 18, and Fig. 19 present examples where LLAMA-3.1-405B errs due to a wrong execution on an Audio-8-bit, a wrong execution on a synthetic sequence, repetition on an Audio-8-bit sequence, and conversion from text to bytes for a text sequence.

| Input Sequence | Generated Program | Produced Sequence |
|---|---|---|
| [249, 252, 254, 255, 254, 249, 246, 250, 254, 255, 255, 253, 252, 254, 255, 0, 1, 1, 3, 5, 3, 1, 1, 4, 6, 6, 4, 1, 0, 1, 3, 3, 0, 254, 255, 255, 254, 254, 253, 253, 254, 254, 253, 253, 253, 255, 1, 0, 255, 255, 255, 0, 2, 2, 1, 1, 0, 1, 1, 0, 1, 1, 0, 0, 255, 254, 254, 255, 255, 254, 253, 252, 254, 254, 255, 255, 254, 253, 253, 255, 254, 255, 255, 255, 255, 255, 254, 255, 0, 0, 2, 4, 5, 4, 2, 1, 2, 5, 6, 6, 3, 1, 0, 1, 2, 1, 1, 0, 255, 254, 253, 253, 254, 255, 255, 254, 251, 251, 253, 255, 0, 255, 255, 254, 254, 255, 1, 2] | ```
a = [249, 252, 254, 255]
b = [254, 249, 246, 250, 254,
     255, 255, 253, 252, 254, 255]
c = [0, 1, 1, 3, 5, 3, 1, 1, 4,
     6, 6, 4, 1, 0, 1, 3, 3, 0]
d = [254, 255, 255, 254, 254,
     253, 253, 254, 254, 253, 253,
     253, 255]
e = [1, 0, 255, 255, 255, 0, 2,
     2, 1, 1, 0, 1, 1, 0, 1, 1, 0,
     0]
f = [255, 254, 254, 255, 255,
     254, 253, 252, 254, 254, 255,
     255, 254]
g = [253, 253, 255, 254, 255,
     255, 255, 255, 255, 254, 255]
h = [0, 0, 2, 4, 5, 4, 2, 1, 2,
     5, 6, 6, 3, 1, 0, 1, 2, 1, 1,
     0]
i = [255, 254, 253, 253, 254,
     255, 255, 254, 251, 251, 253,
     255]
j = [0, 255, 255, 254, 254, 255,
     1, 2]
output = a + b + c + d + e + f +
     g + h + i + j
``` | [249, 252, 254, 255, 254, 249, 246, 250, 254, 255, 255, 253, 252, 254, 255, 0, 1, 1, 3, 5, 3, 1, 1, 4, 6, 6, 4, 1, 0, 1, 3, 3, 0, 254, 255, 255, 254, 254, 253, 253, 254, 254, 253, 253, 253, 255, 1, 0, 255, 255, 255, 0, 2, 2, 1, 1, 0, 1, 1, 0, 1, 1, 0, 0, 255, 254, 254, 255, 255, 254, 253, 252, 254, 254, 255, 255, 254, 253, 253, 255, 254, 255, 255, 255, 255, 255, 254, 255, 0, 0, 2, 4, 5, 4, 2, 1, 2, 5, 6, 6, 3, 1, 0, 1, 2, 1, 1, 0, 255, 254, 253, 253, 254, 255, 255, 254, 251, 251, 253, 255, 0, 255, 255, 254, 254, 255, 1, 2] |

Figure 13: An example generation for LLAMA-3.1-405B on Audio-8-bit where the model succeeds by splitting the sequence to sub-sequences that are later concatenated.

| Input Sequence | Generated Program | Produced Sequence |
|---|---|---|
| [250, 250, 251, 251, 251, 250, 248, 246, 244, 244, 243, 244, 245, 247, 249, 251, 251, 250, 248, 246, 243, 242, 241, 241, 241, 241, 242, 243, 244, 245, 245, 246, 246, 245, 245, 244, 244, 245, 245, 245, 246, 245, 246, 245, 244, 243, 243, 244, 244, 245, 246, 247, 249, 255, 16, 34, 43, 40, 29, 27, 29, 23, 9, 252, 248, 250, 248, 245, 250, 4, 11, 7, 2, 4, 12, 14, 9, 4, 8, 16, 21, 22, 26, 31, 30, 22, 12, 7, 4, 252, 241, 234, 235, 239, 241, 243, 249, 255, 2, 1, 1, 4, 7, 6, 4, 5, 9, 13, 13, 11, 9, 6, 1, 249, 243, 241, 239, 236, 235, 237, 242, 248, 252, 255, 2, 4, 5, 5] | ```a = 250b = [a, a]c = [251, 251, 251]b.extend(c)d = [a, 248, 246, 244, 244, 243,    244, 245, 247, 249]b.extend(d)e = [251, 251, a, 248, 246, 243,    242, 241, 241, 241,    241, 242, 243, 244, 245,       245, 246, 246, 245,    245, 244, 244, 245, 245,       245, 246, 245, 246,    245, 244, 243, 243, 244,       244, 245, 246, 247,    249, 255, 16, 34, 43, 40,       29, 27, 29, 23, 9]b.extend(e)f = [252, 248, 250, 248, 245,    250, 4, 11, 7, 2, 4,       12, 14, 9, 4, 8, 16, 21, 22,          26, 31, 30, 22,    12, 7, 4]b.extend(f)g = [252, 241, 234, 235, 239,    241, 243, 249, 255, 2,    1, 1, 4, 7, 6, 4, 5, 9, 13,       13, 11, 9, 6, 1]b.extend(g)h = [249, 243, 241, 239, 236,    235, 237, 242, 248,    252, 255, 2, 4, 5, 5]b.extend(h)output = b``` | [250, 250, 251, 251, 251, 250, 248, 246, 244, 244, 243, 244, 245, 247, 249, 251, 251, 250, 248, 246, 243, 242, 241, 241, 241, 241, 242, 243, 244, 245, 245, 246, 246, 245, 245, 244, 244, 245, 245, 245, 246, 245, 246, 245, 244, 243, 243, 244, 244, 245, 246, 247, 249, 255, 16, 34, 43, 40, 29, 27, 29, 23, 9, 252, 248, 250, 248, 245, 250, 4, 11, 7, 2, 4, 12, 14, 9, 4, 8, 16, 21, 22, 26, 31, 30, 22, 12, 7, 4, 252, 241, 234, 235, 239, 241, 243, 249, 255, 2, 1, 1, 4, 7, 6, 4, 5, 9, 13, 13, 11, 9, 6, 1, 249, 243, 241, 239, 236, 235, 237, 242, 248, 252, 255, 2, 4, 5, 5] |

Figure 14: An example generation from LLAMA-3.1-405B for Audio-8-bit where the program repeats the input sequence in a verbose way, i.e., by extracting the number 255 to a variable and splitting to sub-sequences.

| Input Sequence | Generated Program | Produced Sequence |
|---|---|---|
| [196, 197, 198, 199, 200, 201, 202, 203, 204, 205, 206, 207, 208, 209, 210, 211, 212, 213, 80, 80, 80, 80, 80, 80, 80, 26, 26, 26, 26, 26, 108, 97, 19, 251, 31, 45, 93, 136, 249, 32, 162, 7, 184, 244, 191, 178, 179, 180, 181, 182, 183, 184, 185, 186, 187, 188, 189, 190, 191, 192, 193, 194, 195, 196, 197, 198, 199, 200, 201, 80, 80, 80, 80, 80, 80, 80] | ```output = (    list(range(196, 214))    + [80] * 7    + [26] * 5    + [108, 97, 19, 251, 31, 45,       93, 136, 249, 32, 162, 7,       184, 244, 191]    + list(range(178, 202))    + [80] * 7)``` | [196, 197, 198, 199, 200, 201, 202, 203, 204, 205, 206, 207, 208, 209, 210, 211, 212, 213, 80, 80, 80, 80, 80, 80, 80, 26, 26, 26, 26, 26, 108, 97, 19, 251, 31, 45, 93, 136, 249, 32, 162, 7, 184, 244, 191, 178, 179, 180, 181, 182, 183, 184, 185, 186, 187, 188, 189, 190, 191, 192, 193, 194, 195, 196, 197, 198, 199, 200, 201, 80, 80, 80, 80, 80, 80, 80] |

Figure 15: An example synthetic sequence where LLAMA-3.1-405B succeeds in leveraging patterns in the input sequence, i.e., repetitions and an increasing range, to generate a correct program.

| Input Sequence | Generated Program | Produced Sequence |
|---|---|---|
| [6, 7, 7, 8, 10, 11, 11, 11, 7, 3, 3, 6, 4, 254, 247, 246, 248, 252, 252, 248, 245, 246, 250, 254, 254, 253, 253, 254, 255, 0, 255, 255, 255, 254, 254, 254, 253, 253, 254, 255, 254, 252, 252, 253, 254, 255, 254, 253, 253, 254, 0, 0, 255, 254, 253, 254, 254, 254, 253, 253, 253, 254, 254, 254, 255, 0, 2, 2, 2, 1, 2, 3, 4, 3, 2, 1, 1, 1, 1, 0, 255, 253, 253, 253, 253, 253, 253, 253, 254, 255, 0, 0, 1, 3, 4, 5, 6, 7, 9, 10, 11, 12, 9, 6, 5, 7, 7, 2, 252, 249, 249, 251, 252, 249, 246, 245, 247, 251, 252, 252, 252, 253, 254, 255, 255, 0, 0, 0] | ```
# Predefined Constants
start_sequence = [6, 7, 7, 8, 10,
    11, 11, 11, 7, 3, 3, 6, 4]
mid_sequence = [255, 0, 255, 255,
    255]
end_sequence = [0, 0, 0]

# Pattern 1: Decreasing by 1
def decreasing_pattern(start,
    length):
    return [start - i for i in
        range(length)]

# Pattern 2: Increasing by 1
def increasing_pattern(start,
    length):
    return [start + i for i in
        range(length)]

# Pattern 3: Repeating a number
def repeat_pattern(num, times):
    return [num] * times

# Generate the Sequence
sequence = start_sequence
sequence += decreasing_pattern
    (254, 5)
sequence += increasing_pattern
    (245, 5)
sequence += repeat_pattern(254,
    3)
sequence += [253, 253, 254, 255]
sequence += mid_sequence
sequence += repeat_pattern(254,
    3)
sequence += decreasing_pattern
    (254, 3)
sequence += [0, 0, 255]
sequence += repeat_pattern(254,
    3)
sequence += decreasing_pattern
    (254, 4)
sequence += [253, 253, 253, 254,
    254, 254, 255, 0]
sequence += [2, 2, 2, 1, 2, 3, 4,
    3, 2, 1, 1, 1, 1, 0]
sequence += mid_sequence
sequence += repeat_pattern(253,
    7)
sequence += [254, 255, 0, 0, 1,
    3, 4, 5, 6, 7, 9, 10, 11, 12,
    9, 6, 5, 7, 7, 2]
sequence += decreasing_pattern
    (252, 3)
sequence += [249, 246, 245, 247,
    251, 252, 252, 252, 253, 254,
    255, 255]
sequence += end_sequence

# Final Output
output = sequence
``` | [6, 7, 7, 8, 10, 11, 11, 11, 7, 3, 3, 6, 4, 254, 253, 252, 251, 250, 245, 246, 247, 248, 249, 254, 254, 254, 253, 253, 254, 255, 255, 0, 255, 255, 255, 254, 254, 254, 254, 253, 252, 0, 0, 255, 254, 254, 254, 254, 253, 252, 251, 253, 253, 253, 254, 254, 254, 255, 0, 2, 2, 2, 1, 2, 3, 4, 3, 2, 1, 1, 1, 1, 0, 255, 0, 255, 255, 255, 253, 253, 253, 253, 253, 253, 253, 254, 255, 0, 0, 1, 3, 4, 5, 6, 7, 9, 10, 11, 12, 9, 6, 5, 7, 7, 2, 252, 251, 250, 249, 246, 245, 247, 251, 252, 252, 252, 253, 254, 255, 255, 0, 0, 0] |

Figure 16: An example program for Audio-8-bit from LLAMA-3.1-405B where the model tries to identify patterns in the input sequence, but the execution does not result in the input.

| Input Sequence | Generated Program | Produced Sequence |
|---|---|---|
| [140, 55, 142, 57, 144, 59, 146, 61, 148, 63, 150, 65, 152, 67, 154, 69, 156, 71, 1, 73, 77, 75, 101, 77, 45, 79, 160, 81, 29, 83, 187, 85, 236, 87, 65, 89, 196, 91, 18, 93, 68, 95, 139, 97, 221, 99, 165, 166, 167, 168, 169, 170, 171, 172, 173, 174, 175, 176, 55, 57, 59, 61, 63, 65, 67, 69, 71, 73, 75, 77, 79, 81, 83, 85, 87, 89, 91, 93, 95, 97, 99] | `a = 140`
`b = 55`
`c = 2`
`d = [a + i * c for i in range(10)`
`    ]`
`e = [b + i * c for i in range(20)`
`    ]`
`f = [101, 45, 160, 29, 187, 236,`
`    65,`
`    196, 18, 68, 139, 221, 165]`
`g = list(range(166, 177))`
`h = [55 + i * 2 for i in range`
`    (23)]`
`output = d[0:10] + e[0:10] + [1]`
`    + e[10:12]`
`     + f + e[12:15] + g + h` | [140, 142, 144, 146, 148, 150, 152, 154, 156, 158, 55, 57, 59, 61, 63, 65, 67, 69, 71, 73, 1, 75, 77, 101, 45, 160, 29, 187, 236, 65, 196, 18, 68, 139, 221, 165, 79, 81, 83, 166, 167, 168, 169, 170, 171, 172, 173, 174, 175, 176, 55, 57, 59, 61, 63, 65, 67, 69, 71, 73, 75, 77, 79, 81, 83, 85, 87, 89, 91, 93, 95, 97, 99] |

Figure 17: An example program for a synthetic sequence from LLAMA-3.1-405B where the model tries to identify patterns in the input sequence, but the execution does not result in the input.

| Input Sequence | Generated Program | Produced Sequence |
|---|---|---|
| [235, 236, 240, 244, 248, 252, 255, 1, 4, 6, 8, 10, 10, 12, 14, 16, 18, 16, 13, 9, 5, 4, 1, 254, 248, 244, 241, 240, 239, 239, 238, 236, 234, 233, 234, 236, 237, 237, 237, 238, 241, 244, 246, 246, 246, 245, 245, 246, 247, 248, 247, 246, 246, 246, 247, 247, 246, 245, 243, 243, 244, 245, 246, 246, 245, 245, 245, 247, 249, 249, 249, 250, 252, 254, 2, 5, 7, 9, 11, 14, 18, 22, 27, 30, 34, 37, 39, 40, 41, 41, 40, 36, 28, 20, 17, 14, 10, 5, 1, 255, 252, 247, 244, 243, 240, 235, 233, 234, 238, 242, 246, 249, 253, 0, 3, 6, 8, 9, 10, 11, 12, 15, 19, 20, 19, 15, 12] | `a = [235, 236, 240, 244, 248,`
`    252, 255]`
`b = [1, 4, 6, 8, 10]`
`c = [10, 12, 14, 16, 18]`
`d = [16, 13, 9, 5, 4, 1]`
`e = [254, 248, 244, 241, 240,`
`    239]`
`f = [239, 238, 236, 234, 233]`
`g = [234, 236, 237]`
`h = [238, 241, 244, 246]`
`i = [245, 245, 246, 247, 248,`
`    247, 246]`
`j = [243, 243, 244, 245, 246]`
`k = [249, 249, 249, 250, 252,`
`    254]`
`l = [2, 5, 7, 9, 11, 14, 18, 22,`
`    27, 30, 34, 37, 39, 40]`
`m = [41, 41, 40, 36, 28, 20, 17,`
`    14, 10, 5, 1]`
`n = [255, 252, 247, 244, 243,`
`    240, 235]`
`o = [233, 234, 238, 242, 246,`
`    249, 253]`
`p = [0, 3, 6, 8, 9, 10, 11, 12,`
`    15, 19, 20]`
`q = [19, 15, 12]`

`output = a + b + c + c + d + e +`
`    f + g + g + g + h + i + i + j`
`     + j + k + l + m + n + o + p`
`    + q` | [235, 236, 240, 244, 248, 252, 255, 1, 4, 6, 8, 10, 10, 12, 14, 16, 18, 10, 12, 14, 16, 18, 16, 13, 9, 5, 4, 1, 254, 248, 244, 241, 240, 239, 239, 238, 236, 234, 233, 234, 236, 237, 234, 236, 237, 238, 241, 244, 246, 245, 245, 246, 247, 248, 247, 246, 245, 245, 246, 247, 248, 247, 246, 243, 243, 244, 245, 246, 243, 243, 244, 245, 246, 249, 249, 249, 250, 252, 254, 2, 5, 7, 9, 11, 14, 18, 22, 27, 30, 34, 37, 39, 40, 41, 41, 40, 36, 28, 20, 17, 14, 10, 5, 1, 255, 252, 247, 244, 243, 240, 235, 233, 234, 238, 242, 246, 249, 253, 0, 3, 6, 8, 9, 10, 11, 12, 15, 19, 20, 19, 15, 12] |

Figure 18: An example program for Audio-8-bit from LLAMA-3.1-405B where the model tries to repeat the input sequence by breaking it to sub-sequences that are later concatenated but still errs.

| Input Sequence | Generated Program | Produced Sequence |
|---|---|---|
| [32, 102, 111, 114, 99, 101, 115, 32, 105, 110, 32, 116, 104, 101, 32, 69, 97, 115, 116, 32, 116, 111, 32, 100, 101, 97, 108, 32, 119, 105, 116, 104, 46, 32, 87, 101, 101, 107, 115, 32, 108, 97, 116, 101, 114, 32, 74, 111, 104, 110, 115, 116, 111, 110, 32, 119, 111, 117, 108, 100, 32, 100, 101, 102, 121, 32, 74, 101, 102, 102, 101, 114, 115, 111, 110, 32, 68, 97, 118, 105, 115, 32, 97, 110, 100, 32, 115, 117, 114, 114, 101, 110, 100, 101, 114, 32, 104, 105, 115, 32, 102, 111, 114, 99, 101, 115, 32, 116, 111, 32, 83, 104, 101, 114, 109, 97, 110, 46, 32, 79, 102, 32, 99, 111, 117, 114, 115, 101] | <pre>letters = '
    abcdefghijklmnopqrstuvwxyz'
numbers = '0123456789'
sentence = "Forces in the East to
    deal with. Weeks later
    Johnston would defy Jefferson
    Davis and surrender his
    forces to Sherman. Of course"

output = [ord(c.lower()) if c.
    lower() in letters else ord(c
    ) for c in sentence]</pre> | [102, 111, 114, 99, 101, 115, 32, 105, 110, 32, 116, 104, 101, 32, 101, 97, 115, 116, 32, 116, 111, 32, 100, 101, 97, 108, 32, 119, 105, 116, 104, 46, 32, 119, 101, 101, 107, 115, 32, 108, 97, 116, 101, 114, 32, 106, 111, 104, 110, 115, 116, 111, 110, 32, 119, 111, 117, 108, 100, 32, 100, 101, 102, 121, 32, 106, 101, 102, 102, 101, 114, 115, 32, 97, 110, 100, 32, 115, 117, 114, 114, 101, 110, 100, 101, 114, 32, 104, 105, 115, 32, 102, 111, 114, 99, 101, 115, 32, 116, 111, 32, 115, 104, 101, 114, 109, 97, 110, 46, 32, 111, 102, 32, 99, 111, 117, 114, 115, 101] |

Figure 19: An example program for the text modality from LLAMA-3.1-405B where the model tries to produce the sequence by writing a text and converting it to Unicode bytes, but errs slightly (the correct text being `" forces in the East to deal with. Weeks later Johnston would defy Jefferson Davis and surrender his forces to Sherman. Of course"`).

