# OpenReview forum: "The KoLMogorov Test: Compression by Code Generation"
_ICLR.cc/2025/Conference — ICLR 2025 Poster_

### Official Review · Reviewer_1MaY · 2024-10-31

**Soundness:** 3
**Presentation:** 3
**Contribution:** 3
**Rating:** 6
**Confidence:** 4

**Summary:**

This paper proposes a novel aspect to evaluate the LLM: by testing its ability to compress discrete sequence data. The proposed benchmark features several advantages such as unlimited data availability and easy to control difficulty in synthetic data generation. The proposed benchmark can be viewed as a sub-task of code-generation. This paper further trained a few models to outperform current models in this task to explore LLM's edge in data compression.

**Strengths:**

- The studied problem is interesting and challenging for LLMs, and not seriously studied before.

- The proposed benchmark is described in detail, making it easy to understand and reproduce.

- The efforts to train a few LLMs proved that this problem is addressable, which offers more context and helps future researches.

**Weaknesses:**

- Better to provide a few examples on where the LLM's ability starts to drop: for tasks than that threshold, LLMs are able to produce perfect compression programs (and separately for each studied and fine tuned LLMs); for tasks harder than that, LLMs fails even with fine-tuning.

- Better study on the edge of LLM: if more training data are provided, if higher rank LoRA applied, or if better curriculum learning applied, will it be stronger, or stops at current level?

- I believe the LLM could perform better if more domain-specific presets of data are assembled, such as DNS sub-sequences with known patterns and known functionalities, yet this seems to be beyond the scope of this paper.

**Questions:**

See sections above.

---

> ### Author Response · Authors · 2024-11-18
> **Response to reviewer 1MaY**
>
> > Better to provide a few examples on where the LLM's ability starts to drop: for tasks than that threshold, LLMs are able to produce perfect compression programs (and separately for each studied and fine tuned LLMs); for tasks harder than that, LLMs fails even with fine-tuning.
>
> Thank you for this suggestion.We present an analysis of accuracy as a function of sequence and program lengths in §B.3, showing that longer programs and sequences are more challenging for SeqCoder. We will add a thorough discussion regarding accuracy for the different operations in future versions. In preliminary experiments we found that accuracy varies between operators, as also been shown in previous works [1, 2]. This provides further evidence that better CodeLMs are needed for improved performance on KT.
>
> [1] What Makes Math Word Problems Challenging for LLMs?
>
> [2] Can We Count on LLMs? The Fixed-Effect Fallacy and Claims of GPT-4 Capabilities
>
> > Better study on the edge of LLM: if more training data are provided, if higher rank LoRA applied, or if better curriculum learning applied, will it be stronger, or stops at current level?
>
> Thank you for this question. Regarding more training data, we present a learning curve in §B.2, Tab.5. More training data is helpful with diminishing returns, as expected. Additionally, larger models perform better. We agree that curriculum learning for KT can be an exciting direction for future work, especially by modeling KT as an RL problem, and added a discussion about curriculum-learning in our Limitations Section.
>
> > I believe the LLM could perform better if more domain-specific presets of data are assembled, such as DNS sub-sequences with known patterns and known functionalities, yet this seems to be beyond the scope of this paper.
>
> Thank you for this interesting suggestion. We agree that assembling data from multiple domains is an exciting direction for future work, and added a discussion in our Limitations Section in the updated paper.
>
> We thank the reviewer again for their helpful review and constructive feedback and are happy to address any additional concerns during the discussion period.

---

### Official Review · Reviewer_ohAG · 2024-11-04

**Soundness:** 3
**Presentation:** 3
**Contribution:** 3
**Rating:** 5
**Confidence:** 3

**Summary:**

This work presents a Kolmogorov test for code generation models.

Experiments are performed on synthetic data, natural text from Wikipedia, audio sequences from LibriSpeech, and human DNA sequences from the GRCh38 assembly.

Standard compression is compared with LLMs and code generation. Specifically, Gzip and LMiC are used as baselines, and compared with baselines that prompt open-weights (Llama-3.1) and closed-weights (GPT-4o) LLMs to generate the shortest Python program that produces the input, and also open-weight models trained on synthetic text and code.

**Strengths:**

1. The result of the open-weight models trained on synthetic text and code is impressive (84.5% accuracy),
and their compression rate outperforms existing baselines.

2. Compression by code generation may serve as a useful additional scalable benchmark for LLMs.

3. The paper is well-written, and the exposition is clear.

**Weaknesses:**

1. My main concern is that the results do not generalize at all from synthetic to real datasets (0 accuracy).

2. In addition to synthetic data, the data used in this work includes natural text, audio, and human DNA sequences from the GRCh38 assembly represented in FASTA format. In protein structure prediction (PSP), sequences of proteins are commonly represented in FASTA format, and their 3D structure predicted. It would be interesting to see the compression results for protein sequences and their structures.
It would be interesting to extend the real-world datasets to other data types.

3. The synthetic DSL includes 3 initiators x 7 modifies x 2 filters x 3 merger functions. It would be interesting to see this action space extended and understand how sensitive the results are as a function of the action space.

**Questions:**

How sensitive are the results as a function of the action space?

---

> ### Author Response · Authors · 2024-11-18
> **Response to reviewer ohAG**
>
> > My main concern is that the results do not generalize at all from synthetic to real datasets (0 accuracy).
>
> Thank you for raising this concern. Generalization from synthetic to real distributions is a significant challenge which we discuss in our Limitations Section. We are hopeful our work can inspire future research focused on tilting the synthetic distribution towards the real one (e.g., by filtering easy-to-detect synthetic sequences), and better learning from the abundant amounts of real data (e.g., by modeling KT as an RL task).
>
> > In addition to synthetic data, the data used in this work includes natural text, audio, and human DNA sequences from the GRCh38 assembly represented in FASTA format. In protein structure prediction (PSP), sequences of proteins are commonly represented in FASTA format, and their 3D structure predicted. It would be interesting to see the compression results for protein sequences and their structures
>
> Thank you for this suggestion. In addition to our synthetic distribution, our work covers five different modalities. We will be happy to add additional modalities the reviewer thinks necessary, including protein sequences in future versions.
>
> > The synthetic DSL includes 3 initiators x 7 modifies x 2 filters x 3 merger functions. It would be interesting to see this action space extended and understand how sensitive the results are as a function of the action space.
>
> Thank you for this suggestion. We will be happy to add additional operators the reviewer thinks necessary.  However, we note that this is not trivial, and will require retraining our SeqCoder models. Please see our response to Q1 for more details.
>
> > How sensitive are the results as a function of the action space?
>
> Thank you for this interesting question. Some operators are harder for the models to learn than others. We will add a thorough discussion regarding accuracy for the different operations in future versions. In preliminary experiments, we experimented with a simplified DSL that includes only the Concatenate and Interleave operations, which was simpler for our models. To summarize, increasing the action space, especially by introducing “harder” operators, results in sequences that are more challenging to models. We will also open-source our data generation framework to allow easy experimentation for future works.
>
> We thank the reviewer again for their constructive feedback, and are happy the reviewer finds KT to be an “useful additional scalable benchmark for LLMs” and our paper “well-written”. We will also be happy to add additional modalities and operators to our experiments, and will add a discussion regarding accuracy for the different operators in future versions. Nevertheless, as we already experiment with six modalities and an extended DSL, we argue it is unlikely to affect the main findings of our work.
>
> As we addressed all points raised by the reviewer, we are hopeful the reviewer will consider raising their score, and are happy to address any additional concerns the reviewer has during the discussion period.

---

### Official Review · Reviewer_y77n · 2024-11-06

**Soundness:** 3
**Presentation:** 3
**Contribution:** 3
**Rating:** 6
**Confidence:** 5

**Summary:**

The papers proposes a method to evaluate code based language models by using them a compression mechanism to approximately compute the Kolmorogov Complexity (which is uncomputable). The aim of the evaluation procedure is to generate the shortest program which produces a certain data point and then stops. The paper proposes simple experiments to show how to approximately compute the Kolmogorov complexity of datums from LLMs. Experiments in the paper use the state of art closed and open source language models, and show that even the SOTA models struggle on the Kolmogorov test proposed in the paper.

**Strengths:**

- The proposed test is an interesting information theoretic test which can be used to compute the information in data samples relative to a model. If samples have smaller programs, then we know that the LLM has understood the structure of that problem/sequence.
- Table 1 is an interesting figure as it shows that even SoTA models currently struggle on at least some benchmarks like Kolmogorov-Test proposed in the paper.
- The experiments in the paper are intuitive, seem to be easy to follow and replicate.
- Test proposed in the paper is different from the usual benchmarks and can be added to any existing code evaluation pipeline by simply computing the size of the program which generates the desired output from the test cases.

**Weaknesses:**

- The connection between Kolmogorov Complexity and Intelligence is not clear in the submission. If we take the popular code based benchmarks, and compute the size of the programs lets say at pass @ 10, then can we say that models which produce shorter programs are more intelligent?
- When using LLMs on the application side, users are often interested in easy to use/understand programs, will producing shorter programs be useful in this case?
- For the synthetic programs that are constructed as part of the dataset, is there a short proof for why they are the shortest length? Are you using some of the properties of the operations to assure that?
    - Here is an example where the training set may not contain the shortest program, for instance consider the following program:
        - seq_1 = range(0,10,2) seq_2 = range(1,11,2) seq_3 = interleave(seq_1,seq_2) output=seq_3 In this case the shorted program is range(0,11,1) and not the program in the training set.
- Generating the shortest program for any given sequence is NP-Complete so what does line 246 correspond to? Are we trusting the LLM to generate the shortest programs? The baseline seems to be unreliable.
- In terms of the paper writing, it would certainly help if you define the Kolmogorov test or present it as an algorithm. While it’s understandable for people with background, it could be hard to understand for general audience.
- For the Lams in Table 1 especially the Llama models, why was few shot prompt / chain of thoughts reasoning not considered as good baselines? For the synthetic tasks, I believe using CoT with few-shot examples could certainly improve performance. It also seems that the SeqCoder-1.5B model is overfitting to the synthetic task instead of actually being intelligent. Would the claim here be that the 1.5B model is more intelligent than GPT-4o since it has a higher accuracy?

**Questions:**

- Are the Llama models instruction tuned or pre-trained?
- What’s the compression rate for the training data? How many bits can it be compressed to?
- The proposed method seems to be very much model dependent, even on the size of the model? Do you add the size of the model while using your test, so that you can normalize your method across models? Otherwise larger models can very well compress all short programs effectively, thus it would then be a test of how large the LLM is.
- In table 1 what could be the reason for the Llama models to have a low accuracy for 8-bit prediction, while having almost double accuracy for 16-bit case?
- A lot of the analysis seems to be heavily focused on Audio datasets/tasks, is there any particular reason for that? Instead of using existing code based benchmarks?
- Section 1 of this paper by Vitanyi (https://arxiv.org/pdf/0809.2754) has some interesting results on the Kolmogorov Complexity of different objects like, very simple objects, complex objects, random objects etc. I think some inspiration could be used from there to construct interesting examples for evaluating the LLMs, for instances showing if the existing SOTA LLMs can encode simple objects, and then progressively build on complex objects.

---

> ### Author Response · Authors · 2024-11-18
> **Response to reviewer y77n (1/2)**
>
> > The connection between Kolmogorov Complexity and Intelligence is not clear in the submission. If we take the popular code based benchmarks, and compute the size of the programs lets say at pass @ 10, then can we say that models which produce shorter programs are more intelligent?
>
> We thank the reviewer for this interesting question. Compression, generation, and intelligence are closely tied [1, 2], and the ability of LMs to compress data has been recently shown to be linearly correlated with intelligence [3]. Many foundational theories of artificial intelligence are based on compression as well [4, 5]. Nevertheless, this does not mean that models that produce shorter programs for popular code benchmarks are more intelligent, and we do not argue that the Kolmogorov Complexity of programs should be used on these benchmarks, which is an interesting research question for follow-up work. In this work, we are concerned with the ability of CodeLMs to compress sequences of data. We clarified this point in the Limitations Section in the updated version of the paper.
>
> [1] The Hutter Prize
>
> [2] Language Modeling Is Compression
>
> [3] Compression Represents Intelligence Linearly
>
> [4] A Theory of Universal Artificial Intelligence based on Algorithmic Complexity
>
> [5] Machine Super Intelligence
>
> > When using LLMs on the application side, users are often interested in easy to use/understand programs, will producing shorter programs be useful in this case?
>
> There is no theoretical guarantee that shorter programs (programs with lower Kolmogorov Complexity) are easier for humans to understand. While readability of programs is an important aspect of Code Generation, we argue it is out of scope for our work where the main emphasis is on data compression.
>
> We believe that the ability of CodeLMs to generate programs in low-level languages that compress data in ways that are challenging for humans to come up with or even understand can be an advantage rather than a limitation of future AI systems, as this could potentially pave the path to better compressors. We clarified this point in the Limitations Section in the updated version of the paper. We further note, that if one uses a prior over programs that is trained on human code, then compressibility under the prior should result in code that “looks like” human-written code.
>
> > For the synthetic programs that are constructed as part of the dataset, is there a short proof for why they are the shortest length? Are you using some of the properties of the operations to assure that?
>
> Although formal guarantees on being the shortest program are impossible to obtain, even in theory, we made a significant effort to ensure that the target program in our program-sequence data is at least close to optimal by:
> - Removing redundant operations - for example, we do not have a range_down operator to remove redundancy between *reverse(range(x))* and *range_down(x)*.
> - Simplicity bias - when programs generate the same sequence, we only keep the shortest program.
>
> We further discussed this point in §A.2 in the updated version of the paper. While we do not guarantee that our programs are indeed the shortest and do not have an efficient method in mind that would enforce it, we argue that the combination of our efforts and the empirical results are sufficient for the scope of this work.
>
> > Generating the shortest program for any given sequence is NP-Complete so what does line 246 correspond to? Are we trusting the LLM to generate the shortest programs?
>
> Line 246 refers to the description of our prompted baselines, which are prompted to generate the shortest program for a given sequence. As we show in our experiments and analysis this is often not the case. We updated the phrasing of this line in the new version.
>
> > In terms of the paper writing, it would certainly help if you define the Kolmogorov test or present it as an algorithm.
>
> We thank the reviewer for this suggestion and added a formal definition of KT in Alg.2 in §A.3 in the updated paper.
>
> > For the LMs in Table 1 especially the Llama models, why was few shot prompt / chain of thoughts reasoning not considered as good baselines? For the synthetic tasks, I believe using CoT with few-shot examples could certainly improve performance.
>
> We thank the reviewer for this suggestion and will add a CoT baseline (we refer the reviewer to our Response to all Reviewers regarding CoT and our response to reviewer xD3h regarding few-shot baselines).
>
> > It also seems that the SeqCoder-1.5B model is overfitting to the synthetic task instead of actually being intelligent. Would the claim here be that the 1.5B model is more intelligent than GPT-4o since it has a higher accuracy?
>
> We do not claim that SeqCoder-1.5B is more intelligent than GPT4-o, because (a) it saw more relevant training examples, and (b) it does not perform better on natural data. We further clarified this in the updated version.

---

> > ### Author Response · Authors · 2024-11-18
> > **Response to reviewer y77n (2/2)**
> >
> > > Are the Llama models instruction tuned or pre-trained?
> >
> > We use the instruction-tuned Llama-3.1 models. We clarified this in §B.1 in the updated version.
> >
> > > What’s the compression rate for the training data? How many bits can it be compressed to?
> >
> > As we discuss in the Introduction, the Kolmogorov complexity  (optimal compression) for our natural sequences is uncomputable. Hence, we include strong Gzip and LMiC baselines, and an Oracle beeline for our synthetic data. We will be happy to consider any additional baselines the reviewer thinks necessary.
> >
> > > The proposed method seems to be very much model dependent, even on the size of the model? Do you add the size of the model while using your test, so that you can normalize your method across models? Otherwise larger models can very well compress all short programs effectively, thus it would then be a test of how large the LLM is.
> >
> > The size of the model is indeed an important factor. For our LMiC baseline, the model is required during decoding time and we discard the model weights for our calculations as we discuss in Footnote 7 (results for LMiC including model weights were reported in [1]). For the CodeLMs, the weights are only required for encoding, and the compressed encoding does not necessitate the weights for decoding. Nevertheless, larger models require more compute which we do not report in our experiments (as we discuss in our Limitations section), and will be happy to add if the reviewer thinks necessary.
> >
> > [1] Language Modeling Is Compression
> >
> > > In table 1 what could be the reason for the Llama models to have a low accuracy for 8-bit prediction, while having almost double accuracy for 16-bit case?
> >
> > High-quality Audio-16-bit data is more challenging than lower-quality Audio-8-bit to traditional compression methods like Gzip (see Tab.4 in §B.2). A possible reason for the higher Precision for the Llama models on Audio-16-bit is that the higher complexity leads to more repetitions of the input sequences, which in-turn leads to higher Precision (see §5.3 for a discussion about repetitions of input data).
> >
> > > A lot of the analysis seems to be heavily focused on Audio datasets/tasks, is there any particular reason for that? Instead of using existing code based benchmarks?
> >
> > We use Audio-MFCC for several analyses because it is a challenging modality (see Tab.4 in  §B.2), where the MFCC encoding represents the main features of the sound (see §3.2 and §A.1 for more information regarding MFCC). For the effect of input length analysis (Tab.2) we use Audio-8-bit because it is a relatively easy modality for Gzip, hence we expect trends to appear earlier on. Our qualitative analysis in §5 is composed of a uniform sample from all modalities.
> >
> > > Section 1 of this paper by Vitanyi (https://arxiv.org/pdf/0809.2754) has some interesting results on the Kolmogorov Complexity of different objects like, very simple objects, complex objects, random objects etc. I think some inspiration could be used from there to construct interesting examples for evaluating the LLMs, for instances showing if the existing SOTA LLMs can encode simple objects, and then progressively build on complex objects.
> >
> > Thank you for referencing this paper, we agree it is highly relevant to our work, and that exploring varying complexities is an interesting research question. Our work currently explores different difficulties by focusing on six different modalities and various sequence lengths. We agree that a curriculum-learning can be an interesting suggestion for training future models and further discuss this in the Limitations Section in the updated version.
> >
> > We thank the reviewer again for their helpful review and for appreciating the soundness, presentation, and contribution of our work. As we addressed all points raised by the reviewer in the updated version, and will include experiments with a CoT baseline in future versions, we are hopeful the reviewer will consider raising their score. We are happy to address any concerns the reviewer has during the discussion period.

---

> > > ### Comment · Reviewer_y77n · 2024-11-28
> > > **Response to authors**
> > >
> > > I think the paper works on an important direction, and I am satisfied with the response. I have updated the score.

---

> > > > ### Author Response · Authors · 2024-11-29
> > > >
> > > > Dear reviewer,
> > > >
> > > > We are glad to hear that you find the paper addresses an important direction and are satisfied with our response. Thank you again for your valuable feedback and thoughtful review.

---

### Official Review · Reviewer_CMwz · 2024-11-06

**Soundness:** 3
**Presentation:** 3
**Contribution:** 3
**Rating:** 10
**Confidence:** 4

**Summary:**

The authors introduce the KOLMOGOROV-TEST (KT), a novel "compression-as-intelligence" test for code generation in large language models (LLMs). Using KT we can prompt or train LLMs to generate the shortest possible programs that reproduce given data sequences, emulating an ideal form of data compression known as Kolmogorov compression, which is theoretically optimal but incomputable priori. The contribution has several advantages: it provides a myriad supply of problem instances of varying difficulty, uses a robust metric (compression rate), and minimizes risks of pretraining data contamination. In evaluating current models like GPT-4o Omni and LLAMA-3.1-405B across multi-modal data types (audio, text, DNA, and synthetic sequences), the authors find that these models perform poorly on both natural and synthetic data. Also authors showed that while training on synthetic data shows improved compression rates, the gains do not transfer well to real-world data for KT.

**Strengths:**

**Originality**:
The authors present an innovative approach by introducing a "compression-as-intelligence" test (KOLMOGOROV-TEST, or KT) that challenges LLMs to generate the shortest programs across multiple data modalities, a novel and ambitious application of LLMs in program synthesis. The use of a multi-modal framework (including audio, text, DNA, and synthetic data) is particularly original, expanding the scope of LLMs in code generation.

**Quality**:
The authors collected program-sequence pairs for supervised training and evaluation, developing a compositional domain-specific language (DSL) and an automatic data generation framework. This framework enables consistent and reliable data generation while avoiding potential biases through uniform priors, which strengthens the robustness of the study. Additionally, the authors investigated the impact of execution feedback on performance, further enhancing the study's depth.

**Clarity**:
The paper includes a detailed analysis section, providing insights into model performance and limitations. The authors clearly document failure cases, particularly highlighting instances where models fail by attempting to reverse-engineer the data generation process rather than compress it effectively. This transparency improves the readability and interpretability of the results.

**Significance**:
KT sets a new benchmark for evaluating LLMs on program synthesis via compression tasks, offering a reliable metric (compression rate) that is both challenging and robust. The framework holds potential for broader applications and improvements in LLM-based code generation by identifying key areas for enhancement, such as in handling complex, real-world data compression.

**Weaknesses:**

**Limited Focus on Code-Specific LLMs**:
Although the paper introduces a compression benchmark specifically for code-generating language models (CODE LMs), the experiments primarily use general-purpose LLMs. Testing specialized CODE LMs could offer a more accurate assessment of model performance for program synthesis tasks, as these models are often fine-tuned with code-specific vocabularies and architectures. Incorporating CODE LMs in future work would enhance the relevance and impact of the findings.

**Understudied Tokenization Impact**:
The impact of tokenization on model performance is underexplored. Prior work (e.g., Delétang et al.) suggests that larger vocabularies may offer more expressive flexibility but can also complicate next-token prediction. Evaluating different tokenization strategies would shed light on their influence on compression rates and model accuracy, potentially guiding optimizations in vocabulary selection to balance expressiveness and predictive ease.

**Suggestions to test Length Generalization**:
Scaling to longer sequences remains an open question. Applying techniques like Rotary Position Embeddings (ROPE) for scaling might provide insights into the model's capacity for length generalization. Additional experiments with longer sequences could clarify how well the model generalizes across variable input sizes, which is critical for practical applications of program synthesis.

**Limited Prompt Optimization**:
The paper could benefit from more work on prompt optimization, as prompt quality can significantly impact model output. Experimenting with prompt tuning strategies or context-enhancing techniques could improve model performance and compression efficiency. This would help achieve the paper’s goal of generating shorter, more precise programs, as optimized prompts can guide the model more effectively toward compact solutions.

**Questions:**

1. **Live Benchmark Availability**:
   Are there any plans to release KT as a live benchmark in addition to the dataset? A live benchmark could provide ongoing insights into model performance and facilitate community-driven improvements over time.

2. **Use of Code-Specific LLMs**:
   Why weren’t code-specific LLMs, such as CodeLLAMA, DeepSeek Coder, or StarCoder, tested in this work? These models have vocabularies optimized for code, which may affect compression rates and generalization performance. Testing code-specific LLMs could provide better priors for transfer learning and potentially enhance performance on program generation tasks.

3. **Training Details**:
   How many epochs or steps were the models trained on for this task? Understanding the training duration would clarify whether models had sufficient exposure to the data to learn effective compression strategies.

4. **Sequence Length and Token Count**:
   The input sequence lengths vary from 16 to 1024 bytes, but could you clarify the number of tokens these represent? This information would help assess how sequence length impacts model performance and how the model tokenizes different sequence lengths.

5. **Data Type Configuration for LLAMA-3.1-405B**:
   Which data type was used for LLAMA-3.1-405B (e.g., bf16, fp16, fp8)? Were any specific data types recommended by the model’s authors tested to ensure compatibility with the architecture? Correct data type configurations can be critical for performance, and confirming this would add clarity to the experimental setup.

---

> ### Author Response · Authors · 2024-11-18
> **Response to reviewer CMwz**
>
> > Limited Focus on Code-Specific LLMs
>
> Thank you for this suggestion. In the paper, we experiment with two model families (Llama-3.1 and GPT4-o) and various sizes for the Llama model. While Llama-2 had an official code-specific variant, termed CodeLlama [1], the official Llama-3 models already perform well on code tasks, e.g., Llama-405B and GPT4-o reach a performance of 88.6 and 87.8 on MBPP Eval Plus, respectively [2].  We will be happy to experiment with additional models the reviewer thinks necessary, including the recently released OpenAI-o1 as suggested by reviewer xD3h.
>
> [1] Code Llama: Open Foundation Models for Code
>
> [2] The Llama 3 Herd of Models
>
> > Understudied Tokenization Impact
>
> Thank you for this suggestion. We agree that tokenization is an important factor that can further improve performance. However, as we use strong CodeLMs that have been shown to perform well on coding tasks, we argue this is out of scope for our current work. Based on the reviewer’s comment, we added a discussion about tokenization in our Limitations section.
>
> > Suggestions to test Length Generalization
>
> Thank you for this interesting suggestion, and please see our response to reviewer 5eaV regarding long-context optimizations. Mainly, we argue that the main bottleneck for additional progress on KT is reasoning rather than long-context abilities. Nevertheless, we agree that stronger long-context CodeLMs can perform better and believe it can be an interesting area for future work.
>
> > Limited Prompt Optimization
>
> Thank you for this suggestion. We will add a CoT baseline for future versions. Please see our response to all reviewers about a CoT baseline for more details.
>
> > Live Benchmark Availability
>
> Thank you for verifying, we plan to have a live leaderboard. We clarified this in the Reproducibility section in the updated paper.
>
> > Use of Code-Specific LLMs
>
> Llama-3.1-405B and GPT4-o have high performance on coding benchmarks [1] and perform better or on par with the models mentioned above. We will be happy to consider any models the reviewer thinks necessary including OpenAI-o1 as suggested by reviewer xD3h.
>
> [1] The Llama 3 Herd of Models
>
> > Training Details
>
> The SeqCoder-1.5B and  SeqCoder-8B models were trained for 20K and 10K steps respectively, as presented in Tab.5 in §B.1. We clarified this in §B.2 in the updated version.
>
> > Sequence Length and Token Count
>
> Because our sequences are composed of numbers in range [0, 255], each element is a single token for the Llama models. Additionally, we use a comma as a separator between elements, so a sequence of length 16 is composed of 31 tokens (16 elements and 15 separators). For the LMiC baseline, we do not consider the cost of generating the separator. We further clarified this in §B.2, Footnote.14 in the updated version.
>
> > Data Type Configuration for LLAMA-3.1-405B
>
> We use the default official BF16 configuration from vLLM and will release code to reproduce our experiments.
>
>
> We thank the reviewer again for their helpful reviewer and constructive suggestions and will be happy to address any concerns the reviewer has during the discussion period.

---

> > ### Comment · Reviewer_CMwz · 2024-11-26
> > **Response to authors**
> >
> > Thanks to the authors for the clarifications. I have updated my score to reflect the change.

---

> ### Author Response · Authors · 2024-11-29
>
> Dear reviewer,
>
>
> We are glad to hear our clarifications were helpful and thank you again for your helpful feedback and thoughtful review.

---

### Official Review · Reviewer_uxqq · 2024-11-08

**Soundness:** 3
**Presentation:** 4
**Contribution:** 3
**Rating:** 8
**Confidence:** 4

**Summary:**

This paper presents two contributions to the field of language model (LM) evaluation and training:

* KoLMogorov Test: A proposed test for language models grounded in compression principles with several benefits. Specifically, there is a preponderance of real-world examples (e.g. audio, text, DNA, or any other modality data that can be represented as bit sequence ) that can used to curate an evaluation set of desired degrees of hardness that cannot be otherwise used for fine-tuning LMs - preventing benchmark hacking. Also, this method allows for computationally cheap and reliable automated evaluation that can measure both the generation accuracy and quality, which is traditionally a bottleneck in language model evaluation setups.
* Synthetic instruction set for fine-tuning LMs for the compression task: This work also turns this test into an LM post-training task and provides a framework for generating a synthetic instruction set to enhance LMs' compression capabilities.

Additionally, this work provides interesting insights on LM fine-tuning:
1. Specifically, small LMs extensively fine-tuned on a synthetic instruction set outperform state-of-the-art prompted models and classical baselines (gzip algorithm).
2. that such performance gains by small fine-tuned models on synthetic data do not translate to the performance gains on real-world data

**Strengths:**

The authors have considered datasets of different modalities and degrees of hardness and analyzed state-of-the-art models from both open and closed-source spaces, making a convincing case for KT as a test for code-generation models. This test has the additional benefits of the corresponding evaluation sets not being directly usable for fine-tuning and the preponderance of hard examples in natural data sources.

The various ablations presented in this work comprehensively capture the limited performance of these SOTA LMs on the task of compression, which will be helpful to the broader community.

The framework for synthetic data generation proposed in this work could be used to augment instruction sets towards large-scale post-trainings of LMs aimed at generalized gains.

**Weaknesses:**

Two points that are fundamental to this work -
1. the efficacy of code-generation language models at compression tasks and
2. some of the observations around fine-tuning LM on compression tasks

have already been discussed in prior work, as the authors of this work aptly reference. This limits the novelty of the present work.

**Questions:**

Since compression is posited as a test of [coding] LM intelligence, it will be interesting to examine the other dimensions of an LM's "intelligence" that correlate with its compression efficacy. A potential experimental set-up to this end could involve fixing a base model and measuring the performance of its variants which have been fine-tuned on disparate tasks such as math reasoning, code, etc., on the KoLMogorov Test. If the KoLMogorov Test exhibits a good correlation with the other intelligence dimensions, its benefits may allow it to subsume other benchmarks. KT may also lend itself as a reward function during LM preference optimizations.

The observations in this work when fine-tuning small LMs on synthetic instruction sets for compression tasks - appear to echo other works from the math-reasoning domain, where extensive fine-tuning allows a relatively smaller model to outperform a larger base model (on tasks such as GSM8k), but without any generalized improvements. If the authors see any fine-grained parallels here - this discussion would be a useful contribution to the community.

---

> ### Author Response · Authors · 2024-11-18
> **Response to reviewer uxqq**
>
> > Two points that are fundamental to this work have already been discussed in prior work
>
> While Language Modeling for compression has been previously explored, as far as we are aware we are the first to suggest compression as a code generation task for CodeLMs. We will be happy to reference additional works the reviewer thinks necessary.
>
> > Since compression is posited as a test of [coding] LM intelligence, it will be interesting to examine the other dimensions of an LM's "intelligence" that correlate with its compression efficacy. A potential experimental set-up to this end could involve fixing a base model and measuring the performance of its variants which have been fine-tuned on disparate tasks such as math reasoning, code, etc., on the KoLMogorov Test.
>
> Thank you for this suggestion. In the paper, we experiment with two model families (Llama and GPT4) and various sizes for the Llama model. While Llama-2 had an official code-specific variant, termed CodeLlama [1], the official Llama-3 models already perform well on code [2], and as far as we are aware our Llama-3.1 baselines are competitive with other models in their respective sizes. We will be happy to experiment with additional models the reviewer thinks necessary, including the recently released OpenAI-o1 as suggested by reviewer xD3h.
>
> [1] Code Llama: Open Foundation Models for Code
>
> [2] The Llama 3 Herd of Models
>
> > The observations in this work when fine-tuning small LMs on synthetic instruction sets for compression tasks - appear to echo other works from the math-reasoning domain, where extensive fine-tuning allows a relatively smaller model to outperform a larger base model (on tasks such as GSM8k), but without any generalized improvements.
>
> Thank you for this suggestion, we clarified this point and added references in the Related Work section in the updated paper, and will be happy to reference additional works the reviewer thinks necessary.
>
>
> We thank the reviewer again for their helpful review and constructive feedback and will be happy to address any concerns the reviewer has during the discussion period.

---

> > ### Comment · Reviewer_uxqq · 2024-11-25
> > **Reply to the authors.**
> >
> > Thanks to the authors for the clarifications. These are satisfactory.

---

> > > ### Author Response · Authors · 2024-11-29
> > >
> > > Dear reviewer,
> > >
> > > We are happy to hear our response was satisfactory. Thank you again for your helpful feedback and thoughtful review.

---

### Official Review · Reviewer_xD3h · 2024-11-08

**Soundness:** 3
**Presentation:** 3
**Contribution:** 2
**Rating:** 5
**Confidence:** 4

**Summary:**

The paper talks about compression of data using LLMs via code generation. The smallest possible way to compress a sequence of data called the Kolmogorov compression is uncomputable. Therefore, the paper tests the capability of code generating LLMs to generate a program that can produce the sequence, thus compressing it. They argue that achieving this requires the models to reason, plan, and search for complex patterns within the sequence to enhance compression performance. The paper evaluates state-of-the-art models, such as Llama-3.1-405b and GPT-4-o, in a zero-shot manner on real-world compression data sources like audio, text, and DNA, demonstrating that these models perform poorly compared to the deterministic gzip baseline. Additionally, the authors assess the models on a synthetic benchmark of (sequence, program) pairs created using a custom domain-specific language (DSL). On this synthetic distribution, they trained a code generation model that outperformed gzip with a uniform prior over the custom DSL. However, they also showed that these models don’t generalize to real data. The paper highlights significant differences between the distributions of real and synthetic sequences, with real sequences exhibiting more repetitions. Through ablation studies, the authors further demonstrate that the trained models do not generalize well to longer sequences.

**Strengths:**

The main strengths of the paper include :

- The paper studies the problem of compression as a code generation problem with current code language models. This is quite novel in the context of language models as previous works have primarily used language models for compression through arithmetic coding. This new framing of the task surely requires the model to understand patterns in the sequence and reason about how to segment the sequence into different subsequences.

- The paper proposes a dynamic benchmark comprising of (sequence, program) pairs which test model’s capability at compressing sequences which are generated using a custom domain specific language (DSL) comprising of different functions that can be realized as python functions. The paper shows that codeLMs trained on this synthetic benchmark can outperform deterministic compression methods like gzip when efficient compression priors are applied to generated functions. Notably, zero-shot prompted state-of-the-art models like GPT-4-o and Llama-3.1-504b fail to reproduce input sequence for more than 66% and 45% of the time, respectively. The dynamic nature of the benchmark, due to its dependency on the customizable DSL, prevents existing LLMs from gaming or memorizing it, as it can generate different examples of varying complexity that the model has not previously encountered during training.

- The paper’s evaluation of code generation based compression on various real world data sources like audio, text and DNA is also noteworthy. In the literature of information theory and communication, these domains have been thoroughly studied for different compression based algorithms. Although none of the models discussed in the paper outperformed the gzip baseline for compression, this finding highlights an interesting phenomenon that is widely discussed in the literature: the numerical reasoning and counting limitations of these LLMs.

**Weaknesses:**

The major weaknesses of the paper include :

- Although the benchmark is effectively designed around compression as a code generation problem, the prompts used in the zero shot evaluation of the models are quite open-ended compared to the structured and closed nature of the DSL design that has been used in the synthetic experiments. This leads to an unfair comparison with the synthetic model.  With better prompt-designs, with better in-context examples, inclusion of DSL definitions in the prompt, ReACT prompts with verification in the loop, these SOTA models can achieve good performance. This is particularly relevant given that feedback and execution improved the accuracy of the trained model on audio-based data, and the majority of errors in SOTA models were execution-based.

- While the assessment using real-world data is sensible, expecting the smaller 1.5B/8B model trained on synthetic data to generalize effectively to real-world data is quite ambitious. The distribution of real data significantly differs from synthetic data; the former has more repetitions, while the latter exhibits more complexity in terms of interleaving sequences. The synthetic data used by the authors consists of smaller sequences and smaller length programs whereas real-world data with more repetitions would result in longer generated programs - something the model wasn't trained on, leading to a significant drop in accuracy. This decrease in accuracy for the trained synthetic model could also be attributed to model capacity, as similar models like LLaMa-3.1-8B also show lower accuracy on real-world data.

- Beyond the evaluation and training of models, it is highly unlikely that LLMs can be traditionally and reliably used for compression instead of deterministic methods like gzip. LLMs are inherently probabilistic models and cannot generate lossless compressions without hallucinations or mistakes in a zero-shot setup. The motivation behind designing such a complex benchmark around the Kolmogorov test is unclear. If the focus is on synthetically generated data/benchmark, small 1.5B models trained on 10k-1M perform well on them with some supervised training. However, if the focus is on real data, realizing real-world data compression as a code generation problem that outperforms gzip is very challenging without better base models, improved prompt design, or better synthetic data design.

**Questions:**

For the experiments, the following should have been additionally explored by the authors :

- The authors didn’t report the effect of execution and feedback on the SOTA LLMs. From Table 3, the authors have just presented the ablation on the trained synthetic model.
- The authors didn’t use models like openai-o1 which can reason before emitting any code and have shown to have good performance on code based tasks. It would be interesting how these inference based reasoning models perform on this task ?
- The authors haven’t reported the accuracy and precision of the Seqcoder 8B 1M model on synthetic data. Does starting from a better base model help in accuracy ?

---

> ### Author Response · Authors · 2024-11-18
> **Response to reviewer xD3h (1/2)**
>
> > Although the benchmark is effectively designed around compression as a code generation problem, the prompts used in the zero shot evaluation of the models are quite open-ended compared to the structured and closed nature of the DSL design that has been used in the synthetic experiments.
>
> Thank you for this suggestion. We will add experiments with a CoT baseline to future versions of the paper (please see our response to all reviewers regarding CoT baseline).
>
> We will also be happy to add an in-context learning (ICL) baseline. However, we note that this may not be trivial to implement reliably, as it raises additional questions, mainly (a) which examples to include in the prompt in a way that will not bias the model towards specific operators, and (b) should we prompt the models to use our DSL (like the trained models) or Python programs (like our zero-shot prompted baselines). While we agree that adding examples for SOTA models is likely to improve accuracy for our synthetic sequences and will be an additional contribution, it is unclear it will be helpful for real sequences without annotating domain-specific examples, which can be challenging. We will be happy to explore this in future versions.
>
> We are happy to receive additional suggestions from the reviewer and will be sure to include these experiments in future versions. One of the reasons we are excited about KT is that solving it will likely require many advanced techniques, including the ones you mention, thus driving progress for years to come.
>
> > While the assessment using real-world data is sensible, expecting the smaller 1.5B/8B model trained on synthetic data to generalize effectively to real-world data is quite ambitious.
>
> We agree that generalization from synthetic to real distributions is a major challenge, which we discuss in our Limitations Section. We are hopeful our results can inspire future research in developing methods that tilt the synthetic distribution towards the real one, e.g., by filtering easy to detect synthetic examples or use of RL techniques. While we agree that lower model capacity can limit generalization, our results with two model sizes (1.5B and 8B, see updated Tab.1 in the new version) do not show better generalization for the larger model. Hence, we hypothesize that novel data generation methods and modeling KT as an RL task are exciting directions for future work.
>
> > Beyond the evaluation and training of models, it is highly unlikely that LLMs can be traditionally and reliably used for compression instead of deterministic methods like gzip. If the focus is on real data, realizing real-world data compression as a code generation problem that outperforms gzip is very challenging without better base models, improved prompt design, or better synthetic data design.
>
> The long-term focus of KT is on real data and we clarified this point in our Conclusion Section in the updated version, thanks to the reviewer’s comment. The motivation for the KT benchmark is primarily to provide a very hard challenge for code generation research, requiring code understanding, reasoning and pattern recognition capabilities beyond current models. Secondarily, with sufficient progress and scale, progress on KT may eventually result in competitive compression methods. The probabilistic nature of LLMs is not an impediment, since one can draw multiple samples (or otherwise use test-time compute) to find short and correct programs at encoding time, it is possible to automatically encode prediction errors (so that only code length and not correctness becomes stochastic), and decoding is deterministic.
>
> We would also like to ask the reviewer to defer judgment on whether it is possible to achieve competitive compression results; in the last decade or so, many impossible-seeming tasks (classifying imagenet with a neural network, building chatbots by scaling language models, etc.) had seemed intractable until they were solved. The space of all programs includes any other compression method and so it is not a question of “if” these methods can outperform, but “how” we can reach that goal. Our aim is to focus research effort in this direction using the KT benchmark.

---

> ### Author Response · Authors · 2024-11-18
> **Response to reviewer xD3h (2/2)**
>
> > The authors didn’t report the effect of execution and feedback on the SOTA LLMs. From Table 3, the authors have just presented the ablation on the trained synthetic model.
>
> We experiment with execution feedback only for our SeqCoder models because (a) the models are trained to generate *compositional* programs where each line is executable on its own, and (b) we can easily *train* the models with execution feedback. We agree that experimenting with execution feedback with SOTA CodeLMs is an exciting direction, however it is not trivial to implement reliably and has recently shown to be challenging without additional training [1, 2]. Hence, we believe that methods that provide additional supervision for SOTA CodeLMs is an exciting direction for future research.
>
> [1] What Makes Large Language Models Reason in (Multi-Turn) Code Generation?
>
> [2] NExT: Teaching Large Language Models to Reason about Code Execution
>
>
> > The authors didn’t use models like openai-o1 which can reason before emitting any code and have shown to have good performance on code based tasks. It would be interesting how these inference based reasoning models perform on this task ?
>
> Thank you for this suggestion. OpenAI-o1 was released on 12/9/24, less than a month before the submission deadline on 1/10/24. In addition, it is significantly more expensive than GPT4-o. Nevertheless, we agree with the reviewer that experimenting with o1 can be an interesting addition, and will be happy to add these experiments in future versions of the paper.
>
> > The authors haven’t reported the accuracy and precision of the Seqcoder 8B 1M model on synthetic data. Does starting from a better base model help in accuracy?
>
> Thank you for this question. We added results with SecCoder-8B to Tab.1. Starting from a stronger, better trained model increases accuracy over our synthetic sequences, but does not lead to better generalization to real sequences.
>
> We thank the reviewer again for their helpful and constructive feedback. We will add experiments with CoT and OpenAI-o1 to future versions of the paper. As we clarify above and was also mentioned by the reviewer, KT is a challenging benchmark that can both inspire new research and effectively evaluate CodeLMs. As we addressed all comments raised by the reviewer, we are hopeful the reviewer will consider raising their score, and are happy to address any additional concerns the reviewer has during the discussion period.

---

> ### Comment · Reviewer_xD3h · 2024-11-29
> **Response to authors**
>
> I have read the authors’ response and have chosen to keep my current score.

---

> ### Author Response · Authors · 2024-11-29
>
> Dear reviewer,
>
> Thank you for acknowledging our response and for your valuable feedback.

---

### Official Review · Reviewer_5eaV · 2024-11-08

**Soundness:** 4
**Presentation:** 3
**Contribution:** 2
**Rating:** 8
**Confidence:** 4

**Summary:**

The paper introduces a benchmark designed to evaluate LLM's capabilities in compression as an indicator of intelligence. The Benchmark requires models to generate the shortest possible python program that reproduces an input data sequence, aligning with the concept of Kolmogorov complexity. The authors assess current language models' performance in KT using data from text, audio, and DNA, as well as synthetic data with specific patterns. Results show that models like GPT4-O and LLAMA-3.1-405B struggle with both natural and synthetic data. The authors also develop a Domain Specific Language (DSL) to create program-sequence pairs for supervised training, leading to trained models that achieve lower compression rates than traditional methods on synthetic data but perform poorly on real data. Authors have conducted further experiments and comparisons with different baselines including GZIP, "Language Model is Compression", etc. The paper suggests that further innovations are necessary for models to generalize well on real-world data within the KT framework.

**Strengths:**

1. Extensive amounts of experiments conducted and insights gathered from it are the major strengths of this paper. Experiments related to trained SeqCoder are  particularly helpful in highlighting the generalization issues and how much it could be overcome with synthetic data.
2. Clear articulation of the DSL Language and of the techniques used in generating the data. The DSL Language and its complexity seems sufficient for benchmarking SOTA LLMs.
3. Easy to read, with key pieces of information and additional analysis generally emphasized appropriately.
4. Setting aside the question of whether LLMs will be widely used to generate code to compress data, this benchmark highlights current limitations of SOTA LLMs, even with basic patterns such as repetition.

**Weaknesses:**

1. Though the breadth of the experiments is extensive, the Paper could have included experiments and analysis of popular prompting techniques that have been shown to increase the reasoning and coding capabilities of LLMs, such as Chain-of-Thought, Tree-of-Thoughts etc. Given the popularity of these techniques and how prevalent CoT is in all major reasoning benchmarks, including at least one prompting technique would have been useful in the analysis.

2. As the paper's main idea is influenced by the Hutter Prize [1], it would be useful to use any of the compressors in the leaderboard [2] as an additional baseline. Specifically, compressors that use LSTM (such as tensorflow-compress) would make make of an interesting baseline for the benchmark.

[1] - http://prize.hutter1.net

[2] - http://mattmahoney.net/dc/text.html

**Questions:**

See weaknesses for questions

Also, does any of the recent works on lifting the length constraint of LLMs help in Length generalization for SeqCoder? e.g. "Efficient Streaming Language Models with Attention Sinks".

---

> ### Author Response · Authors · 2024-11-18
> **Response to reviewer 5eaV**
>
> > Though the breadth of the experiments is extensive, the Paper could have included experiments and analysis of popular prompting techniques that have been shown to increase the reasoning and coding capabilities of LLMs, such as Chain-of-Thought, Tree-of-Thoughts etc.
>
> Thank you for this suggestion. We will add a CoT baseline to future versions (please see our response to all reviewers regarding a CoT baseline).
>
>
> > As the paper's main idea is influenced by the Hutter Prize [1], it would be useful to use any of the compressors in the leaderboard [2] as an additional baseline.
>
> Thank you for this suggestion. Because we use the same 1GB of Wikipedia data as the Hutter prize we can directly use submissions from their leaderboard as additional baselines for the text modality. We added a clarification regarding state-of-the-art results on the  Hutter prize in the updated version of our work in  §B.2. We plan to create a leaderboard to KT, and will make sure to provide a reference from our leaderboard to the Hutter Prize. We will also be happy to experiment with tensorflow-compress on the other modalities if the reviewer thinks necessary.
>
> > Does any of the recent works on lifting the length constraint of LLMs help in Length generalization for SeqCoder? e.g. "Efficient Streaming Language Models with Attention Sinks".
>
>
> Thank you for this question. Our results in §5.2 show that models struggle on longer sequences, and accuracy is near zero for sequence lengths of 128. Because we experiment with Llama-3.1 models [1] that have been shown to perform well on long-context tasks (for completeness we added results with SeqCoder-8B, which is based on Llama-3.1-8B to Tab.1), we hypothesize that the main bottleneck for better performance on KT is due to reasoning, rather than long-context skills. However, in our analysis in §5.3 we see that current models sometimes fail to even repeat input sequences, suggesting that better attention mechanisms are also needed for good performance.
>
> [1] The Llama 3 Herd of Models
>
>
> We thank the reviewer again for their helpful feedback and are happy to address any additional concerns during the discussion period.

---

### Author Response · Authors · 2024-11-18
**Response to all reviewers**

We thank the reviewers for their thorough reviews and constructive feedback. We are also thankful for their overall positive assessment of our paper.


>Response to all reviewers regarding a CoT baseline


Reviewers 5eaV, xD3h, CMwz noted that they would like to see a chain-of-thought (CoT) baseline. We will be happy to include this baseline in the final paper. We also wanted to add that although CoT is a common method to improve performance, recent work found it can be challenging to apply it reliably for code generation tasks and it can also decrease performance [1,2]. Hence, we do not expect these experiments to affect the main findings of our work, and argue that the Kolmogorov-Test (KT) can be a helpful resource for future CodeLMs, as seemed to be agreed upon by the reviewers. Nevertheless, we will be happy to experiment with a CoT baseline in future versions.

[1] To CoT or not to CoT? Chain-of-thought helps mainly on math and symbolic reasoning

[2] What Makes Large Language Models Reason in (Multi-Turn) Code Generation?


> Updated version of the paper

We uploaded a new version of the paper. The main changes include:
- Addressing comments and suggestions by all reviewers.
- SeqCoder-8B results in Tab.1 based on comments by reviewers xD3h and 5eaV.
- Formal definition of KT in Alg.2 in the appendix, based on comment by reviewer y77n.
- We will also be happy to include experiments with OpenAI-o1 (based on suggestion by reviewer xD3h) and analysis for different operators (based on suggestions by reviewers ohAG and 1MaY) in future versions, although we are unsure we will have these and our experiments with the CoT baseline in time for the discussion period.


We respond in detail to all comments in our individual response to the reviewers and will be happy to address any concerns the reviewers have during the discussion period.

---

### Author Response · Authors · 2024-11-25
**Message to reviewers before end of discussion period**

Dear reviewers,

We wanted to thank you again for your reviews, which were very helpful in improving our work. If possible, we would be very happy to know whether all your concerns have been addressed and to answer any follow-up questions during the time remaining for the discussion period.

---

### Meta-Review · Area_Chair_uBqy · 2024-12-22

**Metareview:**

This paper proposes a benchmark of testing LLM's "compression-as-intelligence" ability in code generation. The simple idea is to require models to generate the shortest possible python program that reproduces an input data sequence, aligning with the concept of Kolmogorov complexity. Experiments in the paper use both state-of-the-art closed (GPT-4-o) and open-source (Llama-3.1-405b) language models, using data from text, audio, and DNA, as well as synthetic data with specific patterns, to show that even the SOTA models struggle with the Kolmogorov test. The authors also develop a Domain Specific Language (DSL) to create program-sequence pairs for supervised training, leading to trained models that achieve lower compression rates than traditional methods on synthetic data but perform poorly on real data. In the end, the proposed test and experiments sheds light on the necessity of further innovations to improve the general LLM abilities.

Most reviewers (6 out of 7) rated the work positively. Weighted by the discussion quality, I tend to side with the collective opinion of all the reviewers and propose to accept the work.

**Additional Comments On Reviewer Discussion:**

This paper had a usual number of 7 reviewers due to lateness of some assigned reviewers and an overcompensation of added emergency reviewers. I first want to acknowledge the outstanding job from the authors of addressing this amount of reviews and maintaining discussions.

During the discussion period, the authors were able to convince 3 reviewers to raise their scores, and had 2 others acknowledge positively to their changes.

2 reviewers who rated marginally (5 and 6) never responded during the discussion period, despite multiple pings from AC.

---

### Decision · Program_Chairs · 2025-01-22

Accept (Poster)